# AdaCtrl: Towards *Adaptive* and *Controllable* Reasoning via Difficulty-Aware Budgeting

**Shijue Huang**[*]                                         *shijue.huang@connect.ust.hk*
*The Hong Kong University of Science and Technology (HKUST)*

**Hongru Wang**[*]                                              *hrwang@ed.ac.uk*
*University of Edinburgh*

**Wanjun Zhong**[*]                                         *wjzhong@connect.ust.hk*
*The Hong Kong University of Science and Technology (HKUST)*

**Zhaochen Su**                                              *zsubf@connect.ust.hk*
*The Hong Kong University of Science and Technology (HKUST)*

**Jiazhan Feng**                                            *fengjiazhan@pku.edu.cn*
*Peking University*

**Bowen Cao**                                               *bwcao@link.cuhk.edu.hk*
*The Chinese University of Hong Kong*

**Yi R. (May) Fung**[†]                                             *yrfung@ust.hk*
*MMSense Lab*
*The Hong Kong University of Science and Technology (HKUST)*

**Reviewed on OpenReview:** *https://openreview.net/forum?id=4J2Ako20V4*

## Abstract

With the advent of test-time scaling, Large Reasoning Models have achieved remarkable performance. However, the reinforcement learning process used to unlock these capabilities often leads to uncontrolled generation length, resulting in substantial computational overhead and unnecessary "overthinking" on simple tasks. Current methods either uniformly minimize reasoning tokens, thereby neglecting the necessity for more intricate reasoning on complex tasks, or employ precise token-level control, which often hinges on accurate difficulty estimation and suffers from unreliable model interpretation for nuanced instructions. To address these limitations, we introduce AdaCtrl, a novel framework that can dynamically adjust its reasoning length based on the model's self-assessed problem difficulty and also allow human-in-the-loop control of the budget to prioritize either efficiency or effectiveness. Specifically, we carefully develop a two-stage training pipeline: 1) *Cold-start fine-tuning stage*, where we first design explicit difficulty-aware tags (e.g., "[Easy]" or "[Hard]") to indicate difficulty of problems, and train the model on a curated dataset to align its reasoning behavior with these difficulty levels; and 2) *Difficulty-aware reinforcement learning stage*, which further refines the model's adaptive reasoning behavior and calibrates its self-assessment of problem difficulty. In this way, AdaCtrl not only empowers the model to adaptively assess the difficulty of problem and adjust reasoning budget allocation, but also enables the user to explicitly control the desired reasoning mode by injecting the specific difficulty-aware tag. Empirical results across four benchmarks show that, compared to different types of

---

[*]Equal contribution.
[†]Corresponding author.

baselines, AdaCtrl effectively balances performance and computational efficiency, leading to performance improvements while dynamically reducing response lengths by up to 90%.[1]

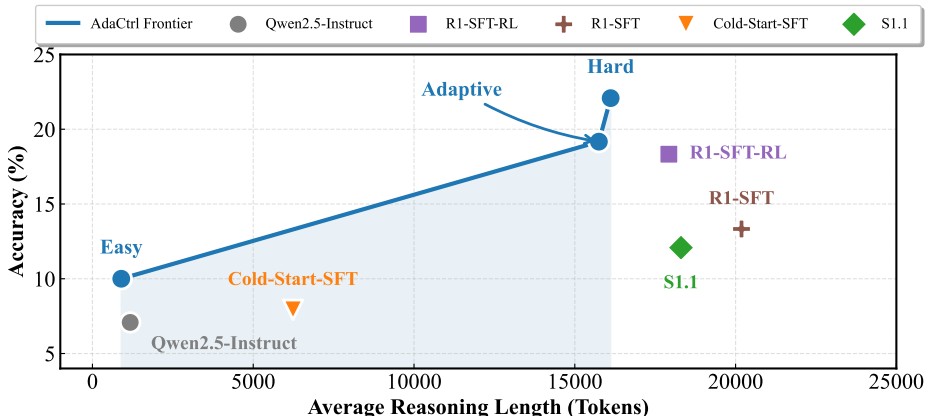

Figure 1: Accuracy-Length Pareto Frontier on AIME 2025. AdaCtrl constructs a dominant frontier (blue line) that sits to the upper-left of all baselines, indicating higher accuracy for a given token budget.

## 1 Introduction

With the emergence of test-time scaling (Snell et al., 2024; Muennighoff et al., 2025; Huang et al., 2025; Balachandran et al., 2025; Zhang et al., 2025; Jiang et al., 2026), large reasoning models such as Deepseek R1 (DeepSeek-AI et al., 2025) and OpenAI o1 (OpenAI, 2024) have demonstrated superior performance across a variety of tasks by thoroughly exploring various reasoning paths prior to generating final answers. However, this capability often comes at a cost: the Reinforcement Learning (RL) optimization used to push reasoning boundaries inevitably leads to an *uncontrolled growth of generation length*. Models tend to discover that generating longer and more redundant reasoning paths can speculatively maximize accuracy rewards (DeepSeek-AI et al., 2025), even when such depth is unnecessary. Consequently, they inevitably introduce significant inference overhead and lead to unnecessary overthinking on simpler problems (Chen et al., 2025; Sui et al., 2025). For instance, even when presented with easy and straightforward questions like "Evaluate $log_2(64)$", these models still tend to engage in lengthy chain-of-thought (Wei et al., 2022), unnecessarily employing advanced meta-reasoning skills such as planning, reflection, and verification (Ryan et al., 2016; Li et al., 2025; Wang et al., 2025a). Such reasoning behavior, while beneficial for complex queries, incurs excessive latency and computational costs, negatively affecting user experience (Wang et al., 2025c).

Recent efforts have explored several ways to improve the reasoning efficiency and mitigate overthinking issue. Some approaches aim to minimize reasoning length across all questions, regardless of their actual complexity (e.g., easy or hard), by enforcing conciseness through fine-tuning (Munkhbat et al., 2025; Ma et al., 2025) or reinforcement learning (Arora & Zanette, 2025; Aggarwal & Welleck, 2025). However, this universal compression strategy may sacrifice reasoning quality on truly complex tasks that require deeper analysis. On the other hand, some methods attempt to precisely control the token budget for each question, seeking a more efficient yet accurate reasoning process (Nayab et al., 2025; Muennighoff et al., 2025; Xu et al., 2025). Nevertheless, such fine-grained control often relies on accurate difficulty assessments and exceptional capabilities of underlying LLMs (e.g., precise instruction-following), which can be inflexible and brittle in practice. It is inherently challenging to determine the optimal token budget beforehand, and models may struggle to reliably interpret or execute fine-grained instructions. These limitations underscore the need for a unified framework that is both *adaptive* and easily *controllable* based on task complexity.

To this end, we propose AdaCtrl, a novel framework to 1) allow the model to dynamically adjust their reasoning effort based on self-assessed difficulty (i.e., adaptive reasoning mode); and 2) enable users to specify

---

[1]Our code is available at: https://github.com/JoeYing1019/AdaCtrl.

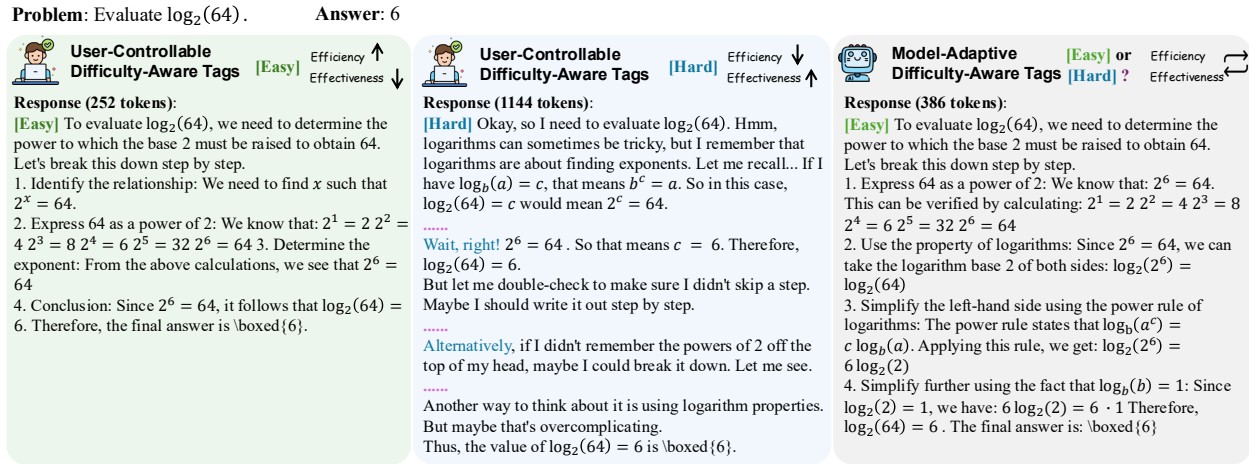

**Problem**: Evaluate $\log_2(64)$. **Answer**: 6

**User-Controllable Difficulty-Aware Tags** [Easy] Efficiency ↑ Effectiveness ↓

**Response (252 tokens)**:

[Easy] To evaluate $\log_2(64)$, we need to determine the power to which the base 2 must be raised to obtain 64. Let's break this down step by step.
1. Identify the relationship: We need to find $x$ such that $2^x = 64$.
2. Express 64 as a power of 2: We know that: $2^1 = 2$ $2^2 = 4$ $2^3 = 8$ $2^4 = 16$ $2^5 = 32$ $2^6 = 64$ 3. Determine the exponent: From the above calculations, we see that $2^6 = 64$
4. Conclusion: Since $2^6 = 64$, it follows that $\log_2(64) = 6$. Therefore, the final answer is \boxed{6}.

**User-Controllable Difficulty-Aware Tags** [Hard] Efficiency ↓ Effectiveness ↑

**Response (1144 tokens)**:

[Hard] Okay, so I need to evaluate $\log_2(64)$. Hmm, logarithms can sometimes be tricky, but I remember that logarithms are about finding exponents. Let me recall... If I have $\log_b(a) = c$, that means $b^c = a$. So in this case, $\log_2(64) = c$ would mean $2^c = 64$.

......

Wait, right! $2^6 = 64$. So that means $c = 6$. Therefore, $\log_2(64) = 6$.
But let me double-check to make sure I didn't skip a step. Maybe I should write it out step by step.

......

Alternatively, if I didn't remember the powers of 2 off the top of my head, maybe I could break it down. Let me see.

......

Another way to think about it is using logarithm properties. But maybe that's overcomplicating.
Thus, the value of $\log_2(64) = 6$ is \boxed{6}.

**Model-Adaptive Difficulty-Aware Tags** [Easy] or [Hard] ? Efficiency ↻ Effectiveness ↩

**Response (386 tokens)**:

[Easy] To evaluate $\log_2(64)$, we need to determine the power to which the base 2 must be raised to obtain 64. Let's break this down step by step.
1. Express 64 as a power of 2: We know that: $2^6 = 64$. This can be verified by calculating: $2^1 = 2$ $2^2 = 4$ $2^3 = 8$ $2^4 = 16$ $2^5 = 32$ $2^6 = 64$
2. Use the property of logarithms: Since $2^6 = 64$, we can take the logarithm base 2 of both sides: $\log_2(2^6) = \log_2(64)$
3. Simplify the left-hand side using the power rule of logarithms: The power rule states that $\log_b(a^c) = c\log_b(a)$. Applying this rule, we get: $\log_2(2^6) = 6\log_2(2)$
4. Simplify further using the fact that $\log_b(b) = 1$: Since $\log_2(2) = 1$, we have: $6\log_2(2) = 6 \cdot 1$ Therefore, $\log_2(64) = 6$. The final answer is: \boxed{6}

Figure 2: Given the same problem, AdaCtrl supports three reasoning modes: the easy mode offers concise answers with less tokens; the hard mode delivers extensive responses with more tokens; and the adaptive mode dynamically allocates reasoning budgets according to the problem complexity.

reasoning mode according to difficulty of problem (i.e., controllable budget allocation). As shown in Figure 2, given a query, AdaCtrl offers three reasoning modes. Two of these modes are manually specified to allow users to control the reasoning budget: (i) Easy, which prioritizes efficiency and provides concise responses to any question; (ii) Hard, which aims for higher effectiveness by elaborating the full reasoning process and delivering detailed information. In addition, an adaptive mode automatically adjusts the reasoning effort based on the complexity of the input query, achieving a balance between effectiveness and efficiency without manual intervention. To achieve this, we start by curating a dataset that covers both *easy* and *hard* subsets and insert corresponding indicator tags (i.e., "[Easy]" or "[Hard]") prior to the model's responses. Then we utilize cold-start fine-tuning to empower the model with foundation capabilities to estimate the complexity of a question and allocate reasoning budgets accordingly, rendering self-aware difficulty estimation. Moreover, we adapt difficulty-aware reinforcement learning framework with carefully designed rewards to calibrate its self-assessment of problem difficulty (i.e., difficulty estimation calibration reward) and to refine the model's adaptive reasoning behavior (i.e., difficulty-aware length reward).

Experimental results on four benchmarks demonstrate that AdaCtrl significantly improves the trade-off between effectiveness and efficiency. It outperforms most baselines across four datasets while efficiently managing the reasoning budget. Specifically, compared with the standard SFT + RL baselines, AdaCtrl achieves accuracy improvements of up to 10.14%, while reducing response length by as much as 91.04%. Further analysis demonstrates that AdaCtrl offers effective human-in-the-loop controllability via explicit difficulty-aware tags, enables accurate difficulty estimation during reinforcement learning, and maintains robust performance under hyperparameter variation. Overall, the contributions are as follows:

- We introduce AdaCtrl, a unified framework for adaptive and controllable reasoning that supports dynamic trade-offs between efficiency and performance, allowing the model to estimate the difficulty of problem and adjust the reasoning mode itself, and also the user to specify the desired reasoning mode to meet diverse needs in practice.

- We present a two-stage training paradigm that integrates cold-start fine-tuning and difficulty-aware reinforcement learning together to foster self-awareness of problem difficulty and supports difficulty-aware budget allocation, considering the differences and dynamics of model capabilities.

- Empirical results on four benchmark datasets demonstrate that AdaCtrl successfully enhance adaptivity and controllability via explicit difficulty-aware tags. Further analysis reveals that AdaCtrl serves as an effective difficulty estimator, and accurately enables difficulty-aware budget allocation.

## 2 Related Work

**Reasoning Efficiency via Supervised Fine-Tuning.** While LLMs achieve impressive performance on complex tasks by generating elaborate multi-step reasoning chains (Dubey et al., 2024; Su et al., 2025), this capability can lead to excessive verbosity and computational overhead for simpler queries. This "overthinking" phenomenon has motivated research into improving reasoning efficiency (Qu et al., 2025a; Wang et al., 2025c; Liu et al., 2025a). One prominent strategy involves Supervised Fine-Tuning (SFT) to guide models towards more concise reasoning. Some SFT works focus on training with inherently shorter reasoning paths. For example models learn adherence to token budgets through specific prompting during data generation (Han et al., 2024). Others distill concise paths from best-of-N sampling (Munkhbat et al., 2025) or fine-tune models to omit intermediate steps for samples where the model is already confident (Yu et al., 2024). Another SFT direction compresses existing reasoning chains. Kang et al. (2024) employ GPT-4 (Achiam et al., 2023) as a compressor then fine-tune a model on these long-to-short CoT mappings. LMskip (Liu et al., 2024) induces step-skipping behavior under step constraints. SPIRIT-FT (Cui et al., 2025) identifies critical reasoning steps using perplexity as a guide for pruning. TokenSkip (Xia et al., 2025) analyzes token importance within CoT outputs for controllable compression. These SFT methods reduce length but often enforce a general conciseness ill-suited for complex problems and typically lack self-assessment of difficulty or user budget control.

**Reasoning Efficiency via Reinforcement Learning.** RL offers another significant avenue for optimizing reasoning efficiency, building upon its success in developing deep reasoning capabilities in models like DeepSeek-Coder (Guo et al., 2025). Many RL approaches incorporate explicit length-based rewards to encourage conciseness alongside accuracy. Some methods link generation length to task difficulty or directives within the prompt: DAST (Shen et al., 2025) adapts CoT length to problem complexity via reward shaping, while LCPO (Aggarwal & Welleck, 2025) controls length using prompt-specified targets. Other techniques normalize length rewards against baselines, as seen in O1-Pruner (Luo et al., 2025), the per-prompt normalization by Arora & Zanette (2025), and the Kimi 1.5 report (Team, 2025). Yeo et al. (2025) proposed a cosine reward to manage length effectively, also highlighting the "length hacking" problem where models artificially extend reasoning. Beyond explicit length rewards, alternative RL strategies include meta-RL for test-time optimization (Qu et al., 2025b), utility maximization for budget awareness (Yu et al., 2025b), preference optimization with heuristics (Chen et al., 2025), and mitigating GRPO's bias towards longer trajectories (Liu et al., 2025b). However, these methods generally lack the explicit user control over reasoning depth offered by AdaCtrl's difficulty-aware tags and do not prioritize training for self-awareness of problem difficulty. Our two-stage SFT-RL framework uniquely addresses these aspects, enabling both autonomous and user-influenced reasoning budgets.

## 3 Method

In this section, we demonstrate the design of our proposed AdaCtrl, which includes: (1) Cold-start fine-tuning that provides initialization for difficulty estimation and difficulty-aware budget adjustment; (2) Difficulty-aware reinforcement learning framework that boosts model's capabilities on response length control and difficulty estimation. The entire framework is demonstrated in Figure 3.

### 3.1 Cold-Start Fine-Tuning

This stage primarily focuses on equipping models with the ability to adhere to output formats that include difficulty-aware tags (e.g., "[Easy]" and "[Hard]")[2] and to control response length accordingly. To curate suitable training data for this purpose, we directly select both easy and hard problems from the DeepMATH dataset (He et al., 2025), which provides difficulty annotations for each problem. For easy problems $\{q_1^e, q_2^e, ..., q_n^e, \}$, we utilize the instruction model, $\mathcal{M}$, to generate concise response, while for hard ones $\{q_1^h, q_2^h, ..., q_m^h, \}$, a strong large reasoning model $\mathcal{R}$ is employed to generate reasoning trace. Then we filter

---

[2]We do not consider more fine-grained category in order to maintain ease of control and ensure reliability.

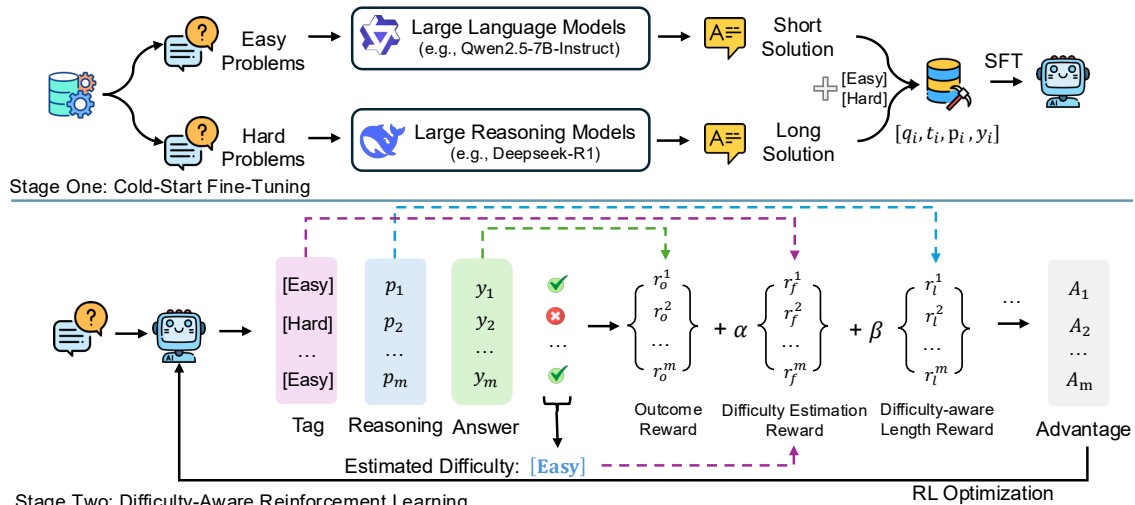

Figure 3: AdaCtrl comprises a two-stage training pipeline: the cold-start finetuning first utilizes both short and long reasoning trajectories to establish basic budget awareness; then a difficulty-aware reinforcement learning framework is utilized to calibrate problem difficulty estimation and develop adaptive reasoning strategies.

the solutions based on answer correctness for both subsets:

$$\mathcal{D}^e = \{(q_i^e, [p_i, y_i]) | [p_i, y_i] = \mathcal{M}(q_i^e), \mathbb{I}(y_i, \hat{y}_i) = 1\}, \tag{1}$$

$$\mathcal{D}^h = \{(q_i^h, [p_i, y_i]) | [p_i, y_i] = \mathcal{R}(q_i^h), \mathbb{I}(y_i, \hat{y}_i) = 1\}, \tag{2}$$

where $[r_i, y_i]$ is the model-generated response that includes reasoning process $p_i$ and the predicted answer $y_i$, $\hat{y}_i$ is the ground truth answer of corresponding samples, and $\mathbb{I}(y_i, \hat{y}_i) = 1$ represents that the model response $[p_i, y_i]$ is correct with the verification of answer $\hat{y}_i$. Then by prepending the "[Easy]" tag for response in $\mathcal{D}^e$ and "[Hard]" tag for those in $\mathcal{D}^h$, we construct a dataset $\mathcal{D} = \{(q_i, [t_i, p_i, y_i])\}$ for cold-start training, where $t_i$ is the difficulty-aware tag. After the cold-start finetuning, the model learns to adhere to the specified format and to generate solutions based on its estimated difficulty of each problem, such as allocating more reasoning tokens for hard problems to enable deeper and more diverse reasoning paths. Notably, this also provides greater controllability for non-expert users, who can easily guide the model's behavior by using simple tags like "[Easy]" and "[Hard]".

## 3.2 Difficulty-Aware Reinforcement Learning

After cold-start fine-tuning, we adapt Group Relative Policy Optimization (GRPO) (Shao et al., 2024) as the reinforcement learning (RL) algorithm because it produces multiple rollouts that can be naturally utilized to estimate difficulty of given question from the perspective of the trained model. Specifically, we carefully design three types of rewards to guide the difficulty-aware reinforcement learning optimization:

**Outcome Accuracy Reward.** The rule-based outcome accuracy reward has been widely utilize in RL training (Guo et al., 2025; Aggarwal & Welleck, 2025; Arora & Zanette, 2025; Yu et al., 2025a; Wang et al., 2025b), which evaluates the correctness of a generated response:

$$r_o(y_i) = \begin{cases} 1.0 & \mathbb{I}(y_i, \hat{y}_i) = 1 \\ -1.0 & \text{otherwise} \end{cases} \tag{3}$$

To ensure more reliable verification, we explicitly require the model to present its final answers in a specified format (i.e., within \boxed{}).

**Difficulty Estimation Calibration Reward.** Accurately estimating problem difficulty is a fundamental aspect of difficulty-aware budgeting. However, since the difficulty of a problem can vary significantly across different models, and obtaining reliable, model-specific difficulty labels is also challenging. To address this, we design a reward function that leverages GRPO's multiple rollouts to estimate a golden difficulty during online training in a natural and effective manner. Specifically, we calibrate the estimated difficulty of the model using the frequency of rollouts that leads to a correct answer. We label a question as easy if the frequency of accurate rollouts exceeds a pre-defined threshold $\delta$, or it is labeled as hard. Moreover, during the rollout process, the model is expected to generate a correct difficulty-aware tag at the beginning of the entire response to match the its capability, so the reward function is designed based on the matches between the generated difficulty-aware tag $t_i$ and estimated tag label $\hat{t}_i$ that is determined by multiple rollouts:

$$r_f(y_i) = \begin{cases} 1.0 & \mathbb{I}(t_i, \hat{t}_i) = 1 \\ 0.0 & \mathbb{I}(t_i, \hat{t}_i) = 0 \\ -1.0 & t_i \text{ cannot be found in } y_i \end{cases} \tag{4}$$

**Difficulty-aware Length Reward.** Different from previous related works that encourage the model to generate concise responses for all problems (Arora & Zanette, 2025; Aggarwal & Welleck, 2025), we hope to encourage such behavior only for easy problems and maintain long thinking capabilities for better tackling hard problems. This helps prevent over-optimization, where the model may fail to engage in deeper thinking when necessary. To prevent unnecessary overthinking, we design a difficulty-aware length reward that encourage concise responses when the generated difficulty-aware tag $t_i$ is "[Easy]":

$$r_l(y_i) = \begin{cases} 1.0 - \frac{1-\cos((l_i^j/L_i)\pi)}{2} & t_i = [\text{Easy}] \\ 0.0 & \text{otherwise} \end{cases} \tag{5}$$

where $l_i^j$ is the $j$-th rollout length for problem $q_i$, and $L_i$ is the max length in the rollout group of problem $q_i$. We leverage the monotonicity of the cosine function within a specific domain (i.e., $0$–$\pi$). This function is easy to tune and provides smooth behavior, as evidenced by a recent study (Yeo et al., 2025). Therefore, the difficulty-aware length reward assigns a lower score as the generated response length increases.

**Overall Reward and Objective.** To enhance the model's adaptive reasoning capabilities and calibrate its self-evaluation of problem difficulty during online training, the overall reward in the reinforcement learning process is computed by integrating the three specifically designed rewards:

$$r(y_i) = r_o(y_i) + \alpha \cdot r_f(y_i) + \beta \cdot r_l(y_i) \tag{6}$$

where $\alpha$ and $\beta$ are hyper-parameters. During optimization, we sample a collection of problems that covers both easy and hard problems from DeepMATH (He et al., 2025) to allow the model to learn dynamic strategies for different types of problems. Following Yu et al. (2025a); He et al. (2025), the policy model $\pi_\theta$ is optimized through the following objective:

$$\mathcal{J}_{\text{GRPO}}(\theta) = \mathbb{E}_{(q,a)\sim\mathcal{D},\{o_i\}_{i=1}^G \sim \pi_{\theta_{\text{old}}}(\cdot|q)} \left[ \frac{1}{G}\sum_{i=1}^G \frac{1}{|o_i|}\sum_{t=1}^{|o_i|} \left( \min\left(\frac{\pi_\theta(o_{i,t} \mid q, o_{i,<t})}{\pi_{\theta_{\text{old}}}(o_{i,t} \mid q, o_{i,<t})}(\theta)\hat{A}_{i,t}, \right. \right. \right.$$
$$\left. \left. \left. \text{clip}\left(\frac{\pi_\theta(o_{i,t} \mid q, o_{i,<t})}{\pi_{\theta_{\text{old}}}(o_{i,t} \mid q, o_{i,<t})}(\theta), 1-\varepsilon, 1+\varepsilon\right)\hat{A}_{i,t}\right) - \beta' D_{\text{KL}}(\pi_\theta||\pi_{\text{ref}}) \right) \right], \tag{7}$$

where $G$ is the group size, $o_i$ is the rollout response, $\hat{A}_{i,t}$ is the advantage of the $i$-th response calculated by normalizing rewards in the group, and $\beta', \varepsilon$ are hyper-parameters. Through this reinforcement learning process, models can more effectively assess problem difficulty relative to their own capabilities and develop adaptive reasoning strategies accordingly.

## 4 Experiment

### 4.1 Experimental Setup

**Model and Datasets.** We adopt Qwen2.5-7B-Instruct and Qwen2.5-14B-Instruct (Qwen et al., 2025) as the backbone model for training. During the cold-start fine-tuning phase, the training problems are drawn from the DeepMATH dataset (He et al., 2025), which assigns each problem a difficulty level ranging from 1 to 9. We categorize problems with difficulty levels of 5 or below as easy, and those above 5 as hard. For easy problems, the backbone model is employed to generate concise responses. In contrast, for hard problems, we incorporate extensive long-form reasoning trajectories generated by Deepseek R1 (DeepSeek-AI et al., 2025). After filtering these trajectories using ground-truth answers, we construct a cold-start SFT dataset comprising 8K instances, which includes 4K with short and 4K with long reasoning chains. To support difficulty-aware reinforcement learning, we further sample an additional 30K examples from DeepMATH that comprise 10K easy and 20K hard problems and are distinct from those in the cold-start fine-tuning dataset.

**Baselines.** To assess the effectiveness of AdaCtrl, we compare it against several baselines that share the same backbone model. These include: (1) Base Model: the unmodified base instruction-tuned model (i.e. Qwen2.5-7B-Instruct, Qwen2.5-14B-Instruct); (2) S1.1: a reasoning-enhanced model obtained by further training the base model on the S1-1.1K dataset (Muennighoff et al., 2025); (3) R1-SFT: a baseline model fine-tuned base model on a curated dataset that includes problems sourced from the cold-start SFT dataset along with all responses generated by Deepseek R1; (4) R1-SFT-RL: a reinforcement learning model that trained (3) with outcome accuracy rewards on the aforementioned 30K RL dataset; (5) Cold-Start: a model fine-tuned based model on the constructed cold-start SFT dataset; (6) Cold-Start-RL: a reinforcement learning model based on (5), trained using outcome accuracy rewards on the same 30K RL dataset. (7) ARM (Wu et al., 2025), a reasoning model capable of adaptively selecting appropriate reasoning formats based on the task at hand.

**Training Details.** For cold-start fine-tuning, we employ the ms-swift framework (Zhao et al., 2024), using a learning rate of 1e-5 and a batch size of 8. For difficulty-aware reinforcement learning, we adopt the VeRL (Sheng et al., 2025) framework, with all RL experiments conducted under a unified setting. Specifically, we follow DAPO (Yu et al., 2025a) to eliminate KL divergence. The policy model is optimized using the AdamW optimizer with a learning rate of 1e-6, a batch size of 256, and a micro-batch size of 32. And we set the value of $\alpha$ and $\beta$ both as 0.5, and the value of $\delta$ as 0.625 during training. During the rollout phase, 16 responses are sampled per prompt, and the maximum generation length is set to 24K tokens. All experiments are conducted on NVIDIA H800 GPUs.

> **Baseline Prompt**
>
> Please reason step by step to answer the following Math Problem, and put your final answer in the format of `\boxed{answer}`.

> **Difficulty-Aware Prompt**
>
> Answer the following math problem, judge the difficulty (Easy/Hard) of given problem and start your response with format: [difficulty here], and put your final answer in the format `\boxed{answer}`.

**Evaluation Settings.** We evaluate our method on four mathematical datasets that span both easy and challenging problems: AIME2024 (Art of Problem Solving, n.d.), AIME2025 (Art of Problem Solving, n.d.), MATH500 (Hendrycks et al., 2021), and GSM8K (Cobbe et al., 2021). The first two datasets consist of more challenging Math Olympiad-style problems, whereas the latter two primarily contain simpler, grade-school level problems, with GSM8K being the easiest among them. Since AIME2024 and AIME2025 each contain only 30 samples, we report the average performance over 8 independent runs for these two datasets. All evaluations are conducted using consistent inference hyper-parameters set to a temperature of 0.7 and a top-p value of 0.8. In this paper, we use a baseline prompt to evaluate all baselines that do not incorporate difficulty-aware

Table 1: Main results of AdaCtrl's adaptive mode. We compute two metrics: Acc.(%) is average accuracy and Len. is the average generated tokens.

| Model | AIME2024 | | AIME2025 | | MATH500 | | GSM8K | |
|---|---|---|---|---|---|---|---|---|
| | Acc.(%)↑ | Len. ↓ | Acc.(%) ↑ | Len. ↓ | Acc.(%) ↑ | Len. ↓ | Acc.(%) ↑ | Len. ↓ |
| *Qwen2.5-7B-Instruct* | | | | | | | | |
| Qwen2.5-7B-Instruct | 11.25 | 1805.60 | 7.08 | 1174.06 | 73.00 | 628.91 | 91.58 | 272.93 |
| S1.1-7B | 16.67 | 19022.60 | 18.33 | 18302.72 | 68.80 | 5824.47 | **91.89** | 1838.12 |
| R1-SFT-7B | 12.08 | 20767.36 | 13.33 | 20184.36 | 62.00 | 8103.41 | 87.34 | 3336.81 |
| Cold-Start-SFT-7B | 11.25 | 5553.00 | 7.92 | 6237.49 | 71.00 | 805.78 | 90.22 | 345.44 |
| ARM-7B | – | – | 16.70 | 3253.00 | 73.90 | 889.00 | 89.20 | 305.00 |
| R1-SFT-RL-7B | 21.25 | 18778.92 | 17.50 | 17924.35 | 66.80 | 8421.57 | 88.93 | 3896.76 |
| Cold-Start-RL-7B | 18.33 | 16911.63 | 14.58 | 15941.76 | 73.00 | 3797.60 | 90.67 | 369.94 |
| **AdaCtrl-7B** | **21.25** | 16889.50 | **19.17** | 15749.08 | **74.00** | 3195.69 | 90.98 | 349.34 |
| *Qwen2.5-14B-Instruct* | | | | | | | | |
| Qwen2.5-14B-Instruct | 11.67 | 1043.20 | 10.42 | 1136.26 | 73.60 | 568.28 | 93.86 | 215.42 |
| S1.1-14B | 32.92 | 17278.03 | **29.58** | 16167.50 | 76.00 | 4393.17 | **95.15** | 1460.26 |
| R1-SFT-14B | 19.58 | 20485.99 | 20.42 | 18447.68 | 65.80 | 6473.98 | 90.67 | 2883.76 |
| Cold-Start-SFT-14B | 12.92 | 4734.60 | 12.50 | 6431.17 | 76.20 | 929.48 | 93.63 | 287.72 |
| ARM-14B | – | – | 20.00 | 3871.00 | 79.10 | 903.00 | 92.50 | 294.00 |
| R1-SFT-RL-14B | 24.17 | 18549.26 | 23.33 | 18294.04 | 70.60 | 6397.75 | 91.05 | 2814.69 |
| Cold-Start-RL-14B | 27.50 | 17722.25 | 24.17 | 17088.44 | 75.60 | 4657.38 | 94.01 | 378.39 |
| **AdaCtrl-14B** | **34.58** | 15173.21 | 25.83 | 14476.93 | **79.20** | 3209.84 | 94.09 | 316.12 |

reasoning. Furthermore, we design a simple difficulty-aware prompt, derived from the baseline prompt, to support our proposed approach, as detailed in Box 1 for baseline prompt and Box 2 for difficulty-aware prompt. To quantify the effectiveness and efficiency of models, we report two metrics: accuracy (Acc.) and the number of tokens generated in the response (Len.).

## 4.2 Main Results

The main results are presented in Table 1. We can have the following observations:

**AdaCtrl Effectively Balances Performance and Reasoning Budget.** AdaCtrl demonstrates competitive overall performance, outperforming most baselines across four datasets while efficiently managing the reasoning budget. Specifically, compared to the RL baseline R1-SFT-RL based on Qwen2.5-7B-Instruct, AdaCtrl-7B achieves comparable accuracy on AIME2024 and further improves accuracy by 1.67% on AIME2025, 7.20% on MATH500, and 2.05% on GSM8K. At the same time, it substantially reduces response lengths by 10.06%, 12.14%, 62.05%, and 91.04% on these datasets, respectively. Under the 14B setting, AdaCtrl-14B increases accuracy by 10.41%, 2.5%, 8.6%, and 3.04% on AIME2024, AIME2025, MATH500, and GSM8K, while compressing the reasoning budget by 18.20%, 20.92%, 49.83%, and 88.77% on the corresponding datasets. Similar trends are also observed when comparing AdaCtrl to the S1.1-7B and S1.1-14B baselines. Furthermore, AdaCtrl consistently surpasses the utility-based ARM baseline, particularly on complex tasks (e.g., AIME2025), by effectively scaling up reasoning length to avoid the capability degradation observed in ARM due to excessive length penalization. Overall, these results suggest that AdaCtrl can adaptively allocate the reasoning budget according to problem difficulty, thereby effectively balancing reasoning efficiency and effectiveness.

**Cold-start Fine-tuning Provides an Effective Foundation for Adaptive Budgeting.** Unlike R1-SFT, which is trained solely on long-form reasoning traces from Deepseek R1, our cold-start fine-tuning strategy (i.e., Cold-Start-SFT) incorporates a combination of both concise and extended reasoning trajectories. This diverse training regimen enables the model to acquire more efficient reasoning strategies, resulting in substantial reductions in response length by 73.26%, 69.10%, 90.06%, and 89.65% for AIME2024, AIME2025, MATH500, and GSM8K, respectively, on the 7B model, and by 76.88%, 54.14%, 85.64%, and 90.02% on

Table 2: Comparison of adaptive and controlled reasoning modes in AdaCtrl. The easy mode reduces reasoning tokens but at the cost of lower performance, making it suitable for fast-response scenarios. Hard mode uses more tokens and yields better results. The adaptive mode automatically estimates problem difficulty to balance effectiveness and efficiency.

| Model | AIME2024 | | AIME2025 | | MATH500 | | GSM8K | |
|---|---|---|---|---|---|---|---|---|
| | Acc.(%)↑ | Len. ↓ | Acc.(%) ↑ | Len. ↓ | Acc.(%) ↑ | Len. ↓ | Acc.(%) ↑ | Len. ↓ |
| AdaCtrl-7B (Adaptive) | 21.25 | 16889.50 | 19.17 | 15749.08 | **74.00** | 3195.69 | 90.98 | 349.34 |
| AdaCtrl-7B (Easy) | 14.58 | 1652.42 | 10.00 | 896.14 | 70.80 | 652.76 | 90.75 | 314.49 |
| AdaCtrl-7B (Hard) | **21.67** | 17562.94 | **22.08** | 16114.87 | 71.20 | 5960.15 | **92.57** | 2058.13 |
| AdaCtrl-14B (Adaptive) | 34.58 | 15173.21 | 25.83 | 14476.93 | **79.20** | 3209.84 | 94.09 | 316.12 |
| AdaCtrl-14B (Easy) | 15.83 | 1394.78 | 11.25 | 985.35 | 75.40 | 578.37 | **94.31** | 284.83 |
| AdaCtrl-14B (Hard) | **36.25** | 15004.55 | **26.25** | 14564.21 | 74.40 | 5321.25 | 93.33 | 2598.28 |

the 14B model. Furthermore, when models initialized from the cold-start checkpoint undergo additional reinforcement learning (i.e., Cold-Start-RL), they demonstrate superior budget control compared to those initialized from R1-SFT (i.e.,R1-SFT-RL), yielding additional reductions in response length of 9.94%, 11.06%, 54.91%, and 91.04% across the same datasets on the 7B model, and 4.46%, 6.61%, 27.20%, and 86.56% on the 14B model. These results underscore the critical role of the cold-start fine-tuning phase in establishing a robust foundation for effective adaptive budgeting in downstream tasks.

**Our Reward Design Enhance Both Reasoning Effectiveness and Efficiency.** Building upon the cold-start fine-tuning model, AdaCtrl outperforms Cold-Start-RL, which applies reinforcement learning based solely on outcome accuracy by achieving notable accuracy improvements of 2.92%, 4.59%, 1.00%, and 0.31% on AIME2024, AIME2025, MATH500, and GSM8K, based on with the 7B model, and improvements of 7.08%, 1.66%, 3.6%, and 0.08% with the 14B model with much less token consumption. These accuracy gains are attributable to our reward function design, which incorporates both difficulty estimation calibration and difficulty-aware length adjustments. By introducing these additional reward signals, our approach enables the model to more accurately assess problem complexity and allocate computational resources accordingly, thereby achieving a more refined balance between reasoning effectiveness and efficiency.

**Human-in-the-Loop Controllability of AdaCtrl.** AdaCtrl introduces difficulty-aware tags that serve as prerequisite tokens during generation, offering explicit control over response length. To assess the controllability of this mechanism, we emulate user intent by specifying either the "[Easy]" or "[Hard]" tag, which correspond to simplified and complex reasoning modes, respectively. As presented in Table 2, the experimental results demonstrate that our approach affords effective control over the reasoning budget. Compared to the adaptive reasoning mode, where the model autonomously determines the problem's difficulty, enforcing the "[Easy]" mode consistently leads to reduced performance across all four datasets. However, it also achieves a substantial decrease in response length by 90.22% and 94.31% on the more challenging AIME2024 and AIME2025 datasets for the 7B model, and by 90.81% and 93.19% for the 14B model, respectively. Conversely, the "[Hard]" mode enhances performance on most datasets and markedly increases response length, with gains of 86.51% and 489.15% on the simpler MATH500 and GSM8K datasets under the 7B setting, and increases of 65.78% and 721.93% under the 14B setting. These findings indicate that AdaCtrl enables precise, human-in-the-loop control over reasoning budgets.

### 4.3 Analysis

**Generalizability across Model Families** To demonstrate the generalizability of AdaCtrl beyond the Qwen architecture, we extended our evaluation to the Llama-3.1-8B-Instruct model. The results, summarized in Table 3, confirm that our proposed two-stage training pipeline effectively transfers to different model families. As shown in Table 3, AdaCtrl consistently maintains its adaptive length control advantages on the Llama architecture. Compared to the R1-SFT-RL baseline, AdaCtrl achieves higher accuracy (e.g., 9.17% vs 7.08% on AIME2024) while significantly reducing token consumption on simpler tasks (e.g., 881.69

Table 3: Performance comparison on Llama-3.1-8B-Instruct. AdaCtrl consistently demonstrates adaptive length control advantages, achieving higher accuracy than baselines while significantly reducing token consumption on simpler tasks, proving the method's strong generalizability.

| Model | AIME2024 | | AIME2025 | | MATH500 | | GSM8K | |
|---|---|---|---|---|---|---|---|---|
| | Acc.(%)↑ | Len. ↓ | Acc.(%) ↑ | Len. ↓ | Acc.(%) ↑ | Len. ↓ | Acc.(%) ↑ | Len. ↓ |
| Llama-3.1-8B-Instruct | 3.58 | 10901.07 | 2.08 | 10383.25 | 46.80 | 4203.35 | 85.14 | 1319.00 |
| + R1-SFT | 3.33 | 19595.04 | 3.33 | 19040.31 | 48.40 | 8792.70 | 83.78 | 3387.50 |
| + Cold-Start-SFT | 2.92 | 15811.69 | 2.08 | 17247.47 | 43.20 | 5327.22 | 83.02 | 641.50 |
| + R1-SFT-RL | 7.08 | 19377.87 | 5.83 | 18026.00 | 53.20 | 11140.40 | 83.09 | 4371.49 |
| + **AdaCtrl** | **9.17** | 16630.41 | **6.25** | 16364.67 | **55.60** | 6397.70 | **87.49** | 881.69 |

Table 4: Hyper-parameter Analysis.$\alpha$ controls the reward weight for difficulty-estimation calibration, while $\beta$ corresponds to the reward weight for the difficulty-aware length objective. The parameter $\delta$ serves as a predefined threshold for distinguishing between easy and hard samples.

| **AdaCtrl-7B** | AIME2024 | | AIME2025 | | MATH500 | | GSM8K | |
|---|---|---|---|---|---|---|---|---|
| | Acc.(%)↑ | Len. ↓ | Acc.(%) ↑ | Len. ↓ | Acc.(%) ↑ | Len. ↓ | Acc.(%) ↑ | Len. ↓ |
| $\alpha = 0.5, \beta = 0.5, \delta = 0.625$ | 21.25 | 16889.50 | 19.17 | 15749.08 | 74.00 | 3195.69 | 90.98 | 349.34 |
| $\alpha = 1.0, \beta = 1.0, \delta = 0.625$ | 23.75 | 14762.78 | 20.00 | 13986.64 | 73.80 | 3134.37 | 90.30 | 325.04 |
| $\alpha = 0.5, \beta = 1.0, \delta = 0.625$ | 22.08 | 15616.19 | 18.75 | 16116.17 | 73.80 | 2836.52 | 91.13 | 324.10 |
| $\alpha = 1.0, \beta = 0.5, \delta = 0.625$ | 21.67 | 17843.66 | 22.08 | 16684.28 | 73.40 | 3727.77 | 91.66 | 365.03 |
| $\alpha = 0.5, \beta = 0.5, \delta = 0.5$ | 22.92 | 13790.53 | 20.42 | 14040.20 | 74.00 | 2888.39 | 90.60 | 388.10 |
| $\alpha = 0.5, \beta = 0.5, \delta = 0.375$ | 20.00 | 13385.01 | 20.00 | 13689.74 | 74.60 | 2181.96 | 90.60 | 323.27 |

vs 4371.49 on GSM8K). Although the absolute performance is constrained by the differing mathematical capabilities of the base model compared to Qwen, these results confirm that our framework effectively enables difficulty-awareness and budget control across different model architectures.

**Hyper-parameter Analysis.** To assess the robustness of our proposed method, we conduct two groups of hyperparameter analyses, as summarized in Table 4. Specifically, we examine: (1) the weights $\alpha$ and $\beta$, which correspond to the difficulty-estimation calibration reward and the difficulty-aware length reward, respectively; and (2) the difficulty threshold $\delta$. For reward weights analysis, we investigate the effect of varying the reward weights across different combinations $\{1 : 0.5 : 0.5, 1 : 1 : 1, 1 : 0.5 : 1, 1 : 1 : 0.5\}$. We observe that accuracy remains largely stable while output length exhibits slight variations. For example, on AIME2024, accuracy stays within 21–23% as the response length ranges from 14K to 17K tokens. Similarly, on GSM8K, accuracy consistently falls between 90–91%, while the response length varies from 325–365 tokens. For difficulty threshold analysis, we further sweep the difficulty threshold $\delta$ over $\{0.625, , 0.5, , 0.375\}$. A smaller value of $\delta$ categorizes more samples as Easy, leading to shorter outputs while maintaining comparable accuracy. For instance, on MATH500, the response length decreases from 3195.69 to 2888.39 and then to 2181.96 tokens, while accuracy remains around 74%. These results collectively demonstrate the robustness of our proposed approach under different hyperparameter settings.

**Ablation Study.** To further assess the effectiveness of the proposed reward functions, we conducted ablation studies by individually removing the difficulty estimation calibration reward $r_f$ and the difficulty-aware length reward $r_l$. As shown in Table 5, the removal of either reward leads to a noticeable decline in performance across all four datasets. These results demonstrate that the combined use of all designed rewards is essential for achieving better optimization.

**AdaCtrl Serves as Good Difficulty Estimator.** To better assess the difficulty estimation capability of AdaCtrl, we analysis the proportion of difficulty-aware tags generated by AdaCtrl-7B across four datasets during reinforcement learning (RL). As illustrated in Figure 4, we first observe that, at the initial stage,

Table 5: Ablation study based on 7B model. $r_f$ is the difficulty-estimation calibration reward and $r_l$ is the difficulty-aware length reward.

| Acc.(%) | AIME2024 | AIME2025 | MATH500 | GSM8K |
|---|---|---|---|---|
| AdaCtrl-7B | **21.25** | **19.17** | **74.00** | **90.98** |
| w/o $r_f$ | 17.08 (-4.17%) | 16.25 (-2.92%) | 72.20 (-1.80%) | 89.92 (1.06%) |
| w/o $r_l$ | 15.42 (-5.83%) | 16.67 (-2.50%) | 68.60 (-5.4%) | 90.60 (-0.38%) |

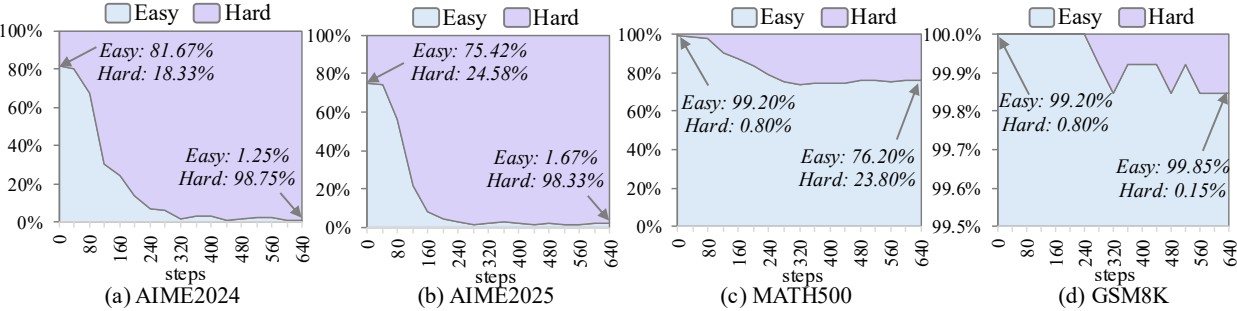

Figure 4: The proportion dynamics of difficulty-aware tags across different datasets during reinforcement learning.

the model tends to classify most samples in all datasets as easy. This is likely because the cold-start SFT primarily teach the model to explicitly generate self-aware difficulty tags in the expected format, rather than to accurately assess problem difficulty from its own perspective. However, following our designed difficulty-aware RL, AdaCtrl predominantly classifies the majority of problems in the AIME2024 and AIME2025 datasets as hard. These datasets consist of challenging math Olympiad-level problems. In contrast, in the MATH500 dataset, which contains a mixture of easy and difficult problems (the majority of which are relatively solvable by current large language models), the model identifies 76.2% of problems as easy. For GSM8K, the simplest dataset among the four, over 99% of problems are categorized as easy, which also accounts for the superior performance of AdaCtrl-14B (Easy) on GSM8K as reported in Table 2. These results align with the actual difficulty levels of the datasets and demonstrate that AdaCtrl develops a robust capability to estimate problem difficulty through RL.

**AdaCtrl Facilitates Accurate Difficulty-Aware Budgeting.** To further investigate the adaptive difficulty-aware budgeting capabilities of AdaCtrl, we analyze AdaCtrl-7B's responses on the MATH500 dataset, which provides difficulty level annotations for each problem. As illustrated in Figure 5 (a), AdaCtrl generates progressively longer and more elaborate responses as the difficulty level increases from 1 to 5, ranging from approximately 0.3K to 6K tokens. This trend indicates that AdaCtrl can accurately regulate its reasoning budget based on its self-assessed estimation of problem difficulty, thereby enabling automatic and adaptive allocation of computational resources.

**Qualitative Analysis of Reasoning Patterns** A notable observation from Table 1 is that AdaCtrl (Hard) generates response lengths approximately 20% shorter than the R1-SFT-RL baseline, despite both being trained on similar long-form reasoning trajectories. This raises the question of whether this reduction comes at the cost of missing critical reasoning steps. Our empirical results negate this concern: on MATH500, AdaCtrl (Hard) outperforms R1-SFT-RL (71.20% vs. 66.80%) while using significantly fewer tokens (5,960 vs. 8,421). We attribute this efficiency to a "conciseness transfer" effect introduced by our mixed-data cold-start strategy, which regularizes the model against non-functional verbosity.

To further investigate the quality of the generated logic, we conducted a fine-grained analysis of reasoning primitives on 100 sampled traces using Gemini-3-Pro. As visualized in Figure 6, AdaCtrl exhibits a richer diversity of reasoning patterns (28 distinct types) compared to the baseline (22 types). While AdaCtrl retains

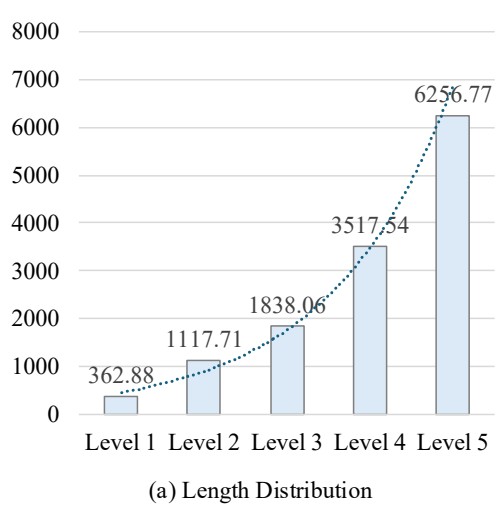
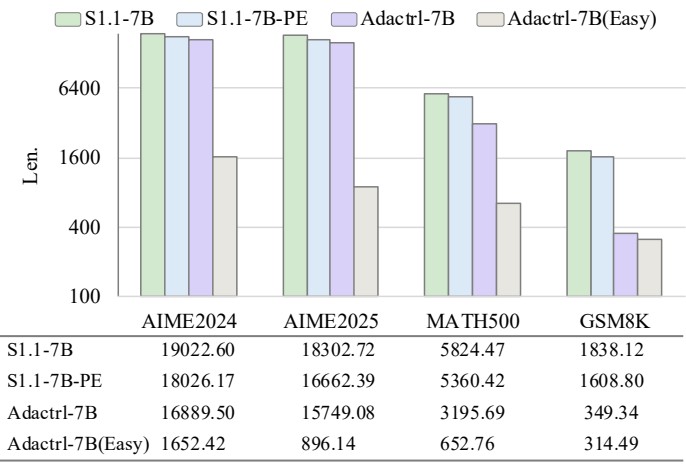

| | AIME2024 | AIME2025 | MATH500 | GSM8K |
|---|---|---|---|---|
| S1.1-7B | 19022.60 | 18302.72 | 5824.47 | 1838.12 |
| S1.1-7B-PE | 18026.17 | 16662.39 | 5360.42 | 1608.80 |
| Adactrl-7B | 16889.50 | 15749.08 | 3195.69 | 349.34 |
| Adactrl-7B(Easy) | 1652.42 | 896.14 | 652.76 | 314.49 |

(a) Length Distribution    (b) Controllability comparison

Figure 5: (a) The length of response in different difficulty levels of problems in MATH500, where higher levels indicate more challenging problems; (b) Controllability comparison of AdaCtrl and prompting-based approach.

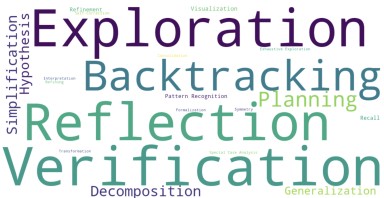
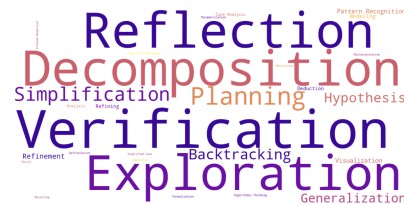

Figure 6: Reasoning Pattern Diversity Analysis. Comparison of reasoning primitives extracted from R1-SFT-RL (Left) and AdaCtrl-Hard (Right).

foundational steps like "Decomposition" and "Reflection", it demonstrates emergent capabilities such as "Algorithmic Thinking", "Heuristic Reasoning", and "Self-Correction" that are less prominent in the baseline. This indicates that AdaCtrl produces denser, higher-quality reasoning chains, effectively "purifying" the logic by eliminating redundant loops while preserving and even enhancing the depth of reasoning.

**Controllability Analysis.** Previous approaches often rely on vanilla prompting methods to achieve token-level control over reasoning budgets. To assess the controllability of AdaCtrl, we compare it with a prompting-based baseline that uses the S1.1-7B model as the backbone and follows Nayab et al. (2025) to augment user prompts with the instruction: "Limit the length of the answer to 500 tokens." The response lengths generated by AdaCtrl in both easy and adaptive modes, as well as those produced by S1.1-7B and its prompting-based budget control variant, are reported in Figure 5 (b). We can observe that, through the prompting-based method (i.e., S1.1-7B-PE) explicitly restricting outputs to 500 tokens, it only yields reductions in response length of only 5.25%, 8.96%, 7.97%, and 12.48% on the AIME2024, AIME2025, MATH500, and GSM8K datasets, respectively, falling significantly short of the targeted 500-token limit. We also observed similar trends when the length constraint was set to 1000 tokens. These results suggest that achieving precise, fine-grained control over output length is challenging due to highly dependent on model's instruction-following capabilities, and that current models may struggle to reliably interpret and execute such granular prompts. In contrast, the easy mode of AdaCtrl achieves substantially greater compression of reasoning budgets, reducing response lengths by 90.22%, 94.32%, 79.57%, and 9.98% across the same datasets. This demonstrates superior controllability, which can be primarily attributed to AdaCtrl's mixed fine-tuning strategy and its design of a difficulty-aware length reward. For a more comprehensive evaluation,

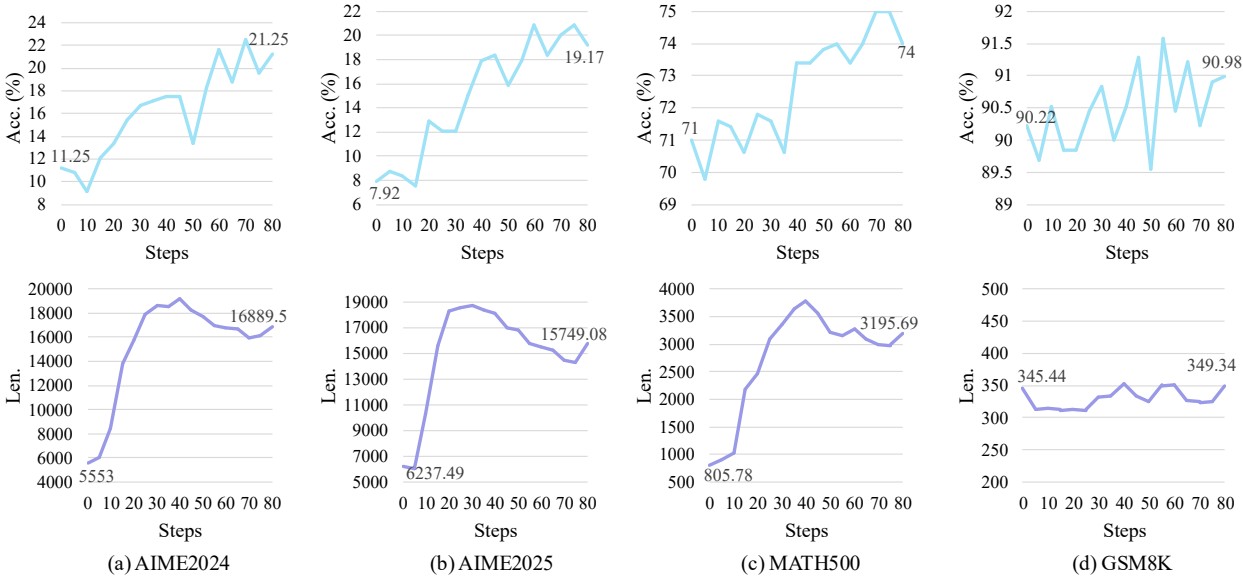

Figure 8: Training dynamics of accuracy and response length across different datasets during reinforcement learning.

we further provide a detailed accuracy-efficiency trade-off comparison between AdaCtrl and prompt-based baselines in Appendix D.

**Length Distribution of Generated Responses.**
To further demonstrate the budget allocation capability of AdaCtrl, we analyze the response length distribution of AdaCtrl-7B across all inference results from the evaluated benchmarks. Specifically, we compute the distributions separately for samples categorized as easy and hard by the model predicted tags. As shown in Figure 7, AdaCtrl clearly differentiates between easy and hard problems. Notably, the response lengths for easy problems are concentrated within a relatively narrow range. These findings indicate that our approach achieves effective and accurate budget control guided by the self-assessed difficulty-aware tags.

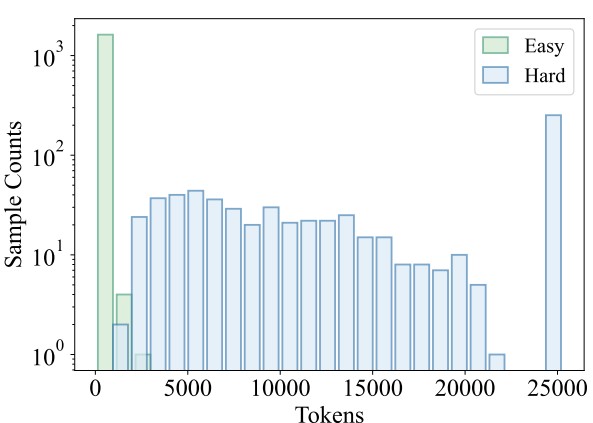

Figure 7: The length distribution of response generated by AdaCtrl-7B.

**Training Dynamics During RL.** To further investigate how the model learns to allocate adaptive reasoning budgets, we analyze both performance trends and budget dynamics of AdaCtrl-7B throughout the reinforcement learning (RL) training process. As illustrated in Figure 8, the model exhibits upward trends in performance across all four datasets, suggesting a progressive enhancement in reasoning capabilities. Regarding budget dynamics, we observe distinct patterns across datasets. On AIME2024, AIME2025, and MATH500, the average response length initially increases rapidly during early training steps, then gradually decreases and stabilizes at a level longer than that before RL training. While for GSM8K, the response length remains relatively stable and close to that observed before RL training.

These findings suggest that the reasoning budget allocation learned during cold-start fine-tuning is insufficient for more complex problems, such as those in AIME2024, AIME2025, and MATH500. Consequently, the model adjusts its budget dynamically in response to actual problem difficulty during RL phase. In contrast, for the comparatively simpler GSM8K dataset, the model is already capable of effectively allocating minimal budgets after cold-start fine-tuning, indicating its ability to distinguish and handle easier problems without requiring significant adjustment.

## 5 Conclusion

In this work, we propose an adaptive and controllable reasoning framework designed to mitigate the problem of overthinking while granting users explicit control over computational resources. To this end, we introduce AdaCtrl that supports both dynamic reasoning budget allocation and user-directed budget adjustments. Our approach utilizes a two-stage training pipeline that combines cold-start fine-tuning with difficulty-aware reinforcement learning. Experiments conducted on four benchmark datasets demonstrate that AdaCtrl effectively allocates reasoning budgets based on self-assessed problem difficulty, leading to performance improvements while dynamically reducing response lengths by 10%–90%. This enables flexible trade-offs between efficiency and performance. Furthermore, AdaCtrl unlocks the potential of human-in-the-loop control towards reasoning budgets according to tailored needs.

### Broader Impact Statement

AdaCtrl enables adaptive and user-controllable allocation of reasoning budgets in large language models, aiming to mitigate unnecessary overthinking while preserving performance on harder queries. By reducing response length when extensive reasoning is not needed, it can lower inference cost and latency, which may also reduce the environmental footprint of large-scale deployment. The difficulty-aware tags provide a lightweight human-in-the-loop mechanism to steer efficiency–effectiveness trade-offs in practical applications. As with most LLM research, there is a general dual-use risk; however, since this work primarily changes how computational effort is allocated rather than enabling new capabilities, we expect the incremental risk introduced by this method to be limited.

### Limitations and Future Work

Several limitations of our study should be noted. First, to keep the control interface lightweight and reliable, AdaCtrl currently uses a binary difficulty schema (i.e., `[Easy]` vs. `[Hard]`), which may be less expressive for intermediate cases. Second, our human-in-the-loop mechanism is instantiated as tag-based control, which provides simple steering but does not yet capture richer user intent such as preferences over verbosity, latency, or explanation style. Third, although we introduce explicit calibration to improve self-assessed difficulty, the estimated difficulty can still be imperfect in some cases, which may lead to occasional over- or under-allocation of reasoning effort.

Future work will explore finer-grained and hierarchical control signals beyond binary tags to enable smoother and more flexible budget allocation while preserving robustness. We will also extend human-in-the-loop control beyond tag-only steering by incorporating richer feedback signals (e.g., preference- or correction-based guidance) to better align budgeting behavior with user intent. Additionally, we plan to further improve difficulty estimation calibration (e.g., by leveraging more informative uncertainty signals from rollouts and training dynamics), strengthening the consistency between self-assessed difficulty and actual solvability.

### Acknowledgments

This research was supported in part by BYD (Grant BYD26EG05), as well as by the Frontier Technology Research for Joint Institutes with Industry Scheme (Grant WEB26EG02).

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

## Appendix

## A    Analysis of Difficulty Estimation Capability

To further quantify the reliability of this difficulty estimation and address concerns regarding potential false negatives (i.e., identifying hard problems as easy), we evaluated the tagging accuracy of AdaCtrl. Ground-truth difficulty was determined by the pass rate of 16 parallel rollouts (problems solved $> 10$ times were labeled Easy). As shown in Table 6, AdaCtrl achieves remarkable precision ($> 92\%$ on AIME and $> 96\%$ on GSM8K) across both sizes. This indicates that the difficulty estimation calibration reward ($r_f$) effectively aligns the model's self-assessment with its actual capabilities, making "first-token commitment" a robust decision mechanism rather than a brittle constraint.

Table 6: Difficulty Tagging Accuracy (%) of AdaCtrl across benchmarks. High accuracy indicates robustness against mode collapse and false negatives.

| Model | AIME2024 | AIME2025 | MATH500 | GSM8K |
|---|---|---|---|---|
| AdaCtrl-7B | 93.33 | 94.58 | 87.82 | 96.58 |
| AdaCtrl-14B | 92.08 | 95.42 | 89.47 | 96.40 |

## B    Detailed Analysis of Performance Trade-offs in Forced Modes

In the main text, we observed that on mixed-difficulty datasets like MATH500, the forced *Hard* mode (71.20%) yields slightly lower accuracy than the *Adaptive* mode (74.00%). To investigate the underlying causes, we conducted a fine-grained error analysis tracking samples where correctness "flipped" between the Adaptive mode (specifically when it autonomously selected the Easy tag) and the forced Hard mode.

Table 7: Error analysis of forcing Hard mode on MATH500. "Gain" denotes samples corrected by switching from Easy to Hard, while "Loss" denotes samples that were correct in Easy mode but failed in Hard mode.

| Model | Gain (Easy $\times \to$ Hard $\checkmark$) | Loss (Easy $\checkmark \to$ Hard $\times$) | Overthinking Analysis within "Loss" |
|---|---|---|---|
| AdaCtrl-7B | 27 samples | 47 samples | **32/47** cases initially derived the correct answer but drifted into error due to forced verbosity. |
| AdaCtrl-14B | 18 samples | 38 samples | **27/38** cases failed due to similar redundancy-induced errors. |

As illustrated in Table 7, the forced Hard mode indeed demonstrates the potential to raise the performance ceiling, evidenced by the 27 'Gain' samples where extended reasoning successfully solved complex problems that failed in the concise mode. However, this benefit is currently outweighed by the 47 'Loss' cases. Crucially, our error analysis reveals that these failures do not stem from a lack of capability. Taking AdaCtrl-7B as an example, human validation confirms that in 32 out of the 47 'Loss' cases (approx. 68%), the model had initially derived the correct answer but was compelled to drift into incorrect revisions due to "forced verbosity." This indicates that the "intrinsic capability" of the Hard mode is actually higher than its final accuracy suggests, but it is compromised by "thinking noise" such as unnecessary self-doubt or hallucinatory verification on simpler queries. While this noise is detrimental on mixed datasets, the dynamic shifts entirely on genuinely challenging benchmarks like AIME, where deep reasoning is indispensable and leads to significant gains (e.g., boosting 7B accuracy to 22.08%). This contrast underscores the critical value of AdaCtrl's Adaptive mode: by autonomously identifying problem difficulty, it intelligently navigates this trade-off, successfully filtering out the overthinking noise on simpler tasks while preserving the capacity to deploy extensive reasoning budgets when truly needed.

Regarding the comparison between the *Easy* mode and the vanilla model, we emphasize that our method maintains comparable efficiency and performance on simple tasks while delivering superior robustness on

challenging ones. On GSM8K, the response length is remarkably similar (314 vs 273 tokens) with competitive accuracy (90.75% vs 91.58%). However, on complex benchmarks where vanilla models typically struggle, our Easy mode actually outperforms the vanilla baseline: for instance, on AIME 2024, it achieves 14.58% accuracy compared to Vanilla's 11.25%, and on AIME 2025, it reaches 10.00% vs 7.08%. This highlights the core value of our framework: it offers the flexibility to secure competitive results with merely 1/10th of the standard reasoning budget (via Easy mode), while retaining the capacity to dynamically expand reasoning length by 10x to unlock maximum performance. This wide dynamic range allows AdaCtrl to navigate the performance-cost trade-off more effectively than any static baseline.

## C   Analysis of Baseline Prompting Fairness

To ensure a rigorous comparison and decouple the effects of prompt engineering from our training methodology, we conducted an additional experiment to assess the fairness of using different prompts for baselines and AdaCtrl. The "Baseline Prompt" used in our main evaluation was modeled after the official system prompt recommended for the Qwen-Math series, ensuring optimal native performance for the baselines.

We applied the "Difficulty-Aware Prompt" (used in AdaCtrl) to the vanilla Qwen2.5-Instruct baseline to determine if performance gains were attributable to the prompt structure itself. As shown in Table 8, replacing the official baseline prompt with the difficulty-aware prompt leads to a noticeable drop in accuracy across most benchmarks. For instance, the 14B model's accuracy on AIME2024 falls from 11.67% to 4.58%. We attribute this degradation to semantic interference, as the vanilla model lacks the internalized alignment to map difficulty tags to appropriate reasoning budgets. Consequently, the extra instructions function as out-of-distribution noise rather than valid control signals. This outcome confirms that the standard "Baseline Prompt" represents the most robust and favorable configuration for the vanilla model, validating the fairness of our original experimental setup.

Table 8: Comparison of Baseline Prompt vs. Difficulty-Aware Prompt on vanilla Qwen2.5-Instruct models. Applying the difficulty-aware prompt to the baseline results in performance degradation, confirming that the baseline prompt is the fairer and stronger setting for comparison.

| Model | AIME2024 | | AIME2025 | | MATH500 | | GSM8K | |
|---|---|---|---|---|---|---|---|---|
| | Acc.(%)↑ | Len. ↓ | Acc.(%) ↑ | Len. ↓ | Acc.(%) ↑ | Len. ↓ | Acc.(%) ↑ | Len. ↓ |
| Qwen2.5-7B-Ins (Baseline Prompt) | 11.25 | 1805.60 | 7.08 | 1174.06 | 73.00 | 628.91 | 91.58 | 272.93 |
| Qwen2.5-7B-Ins (Diff-Aware Prompt) | 10.42 | 1715.62 | 5.83 | 1402.52 | 67.60 | 538.29 | 87.34 | 237.83 |
| Qwen2.5-14B-Ins (Baseline Prompt) | 11.67 | 1043.20 | 10.42 | 1136.26 | 73.60 | 568.28 | 93.86 | 215.42 |
| Qwen2.5-14B-Ins (Diff-Aware Prompt) | 4.58 | 830.68 | 5.00 | 686.01 | 73.00 | 409.86 | 85.75 | 188.47 |

## D   Efficiency-Accuracy Trade-off in Prompt-based Controllability

Table 9: Comparison of accuracy and length between Prompt-based Control (S1.1-7B-PE) and AdaCtrl. The prompt-based method fails to achieve genuine budget control (e.g., >18k tokens on AIME) and underperforms AdaCtrl in efficiency-accuracy ratio.

| Model | AIME2024 | | AIME2025 | | MATH500 | | GSM8K | |
|---|---|---|---|---|---|---|---|---|
| | Acc.(%)↑ | Len. ↓ | Acc.(%) ↑ | Len. ↓ | Acc.(%) ↑ | Len. ↓ | Acc.(%) ↑ | Len. ↓ |
| S1.1-7B-PE | 17.50 | 18026.17 | 16.67 | 16662.39 | 70.80 | 5360.42 | 90.83 | 1608.80 |
| AdaCtrl-7B (Easy) | 14.58 | **1652.42** | 10.00 | **896.14** | 70.80 | **652.76** | 90.75 | **314.49** |
| AdaCtrl-7B (Adaptive) | **21.25** | 16889.50 | **19.17** | 15749.08 | **74.00** | 3195.69 | **90.98** | 349.34 |

In Section 4.3, we demonstrated that prompt-based baselines (e.g., S1.1-7B-PE with instructions to "limit length") fail to achieve effective token reduction. To further assess the trade-off between controllability and

effectiveness, we supplemented the accuracy data for this baseline and compared it directly with AdaCtrl's Easy and Adaptive modes.

As shown in Table 9, relying solely on prompting not only fails in length control but also yields an inferior efficiency-accuracy ratio. For instance, on GSM8K, while S1.1-7B-PE achieves 90.83% accuracy, it consumes ∼1,608 tokens. In contrast, AdaCtrl (Easy) matches this accuracy (90.75%) with only 314 tokens, achieving a **5×** efficiency improvement. On challenging tasks like AIME 2024, although S1.1-7B-PE shows higher accuracy than the Easy mode, its length balloons to over 18,000 tokens, indicating a total failure of budget control. Meanwhile, AdaCtrl (Adaptive) significantly outperforms the prompt baseline (21.25% vs. 17.50%) with reduced token usage, demonstrating a superior Pareto frontier.

