# OpenReview forum: "AdaCtrl: Towards Adaptive and Controllable Reasoning via  Difficulty-Aware Budgeting"
_TMLR — Accepted by TMLR_

### Review · Reviewer_nv4F · 2025-12-23

**Summary Of Contributions:**

**Paper summary**

- Method:
    - They propose a framework that dynamically adjusts the reasoning length based on the model’s self-assessed problem difficulty.
    - their framework consists of two stages:
        - cold start finetuning on curated datasets with explicit difficulty tags. The dataset they used is DEEPMATH that includes difficulty annotations from 1 to 9 which they map to easy (1-5) and hard (6-9).
        - reinforcement learning to calibrate model’s assessment of problem difficulty. They use GRPO and optimize three rewards:
            - outcome accuracy reward
            - difficulty estimation calibration reward
            - difficulty-aware length reward
    - They experiment on Qwen2.5-instruct 7B and 14B as base models.
- Training data:
    - to generate reasoning outputs for training data, the base models are used for easy problems while DeepSeek R1 is used for difficult problems. the finetuning data includes 4K easy and 4K hard instances (cold-start dataset).
    - for RL, they use 10K easy and 20K hard instances for training (RL dataset).
- Baselines:
    - base models: Qwen 2.5-instruct 7B and 14B
    - S1.1: a reasoning model obtained by finetuning the base model on S1-1.1K dataset (no description of the data is provided).
    - R1-SFT: a model obtained by finetuning the base models on a curated dataset similar to the cold-start dataset described before, but with all answers generated using DeepSeek R1.
    - R1-SFT-RL: an RL model obtained by training R1-SFT model on the RL dataset using the outcome accuracy reward.
    - Cold-Start: a finetuned base model on the cold-start dataset.
    - Cold-Start-RL: an RL model obtained by training Cold-Start model on the RL dataset using the outcome accuracy reward.
- Evaluation data:
    - GSM8K (easy)
    - MATH500 (easy)
    - AIME2024 (hard)
    - AIME2025 (hard)
- Results:
    - Compared to the tested baselines, AdaCtrl provides the best balance between performance and reasoning budget.
    - finetuning on their cold-start dataset is better than finetuning on DeepSeek R1 responses only.
    - training with the three rewards achieves better efficiency effectiveness balance than training on the outcome accuracy reward only.
    - AdaCtrl provides user controllability over reasoning budget.
    - AdaCtrl is robust to hyperparameter changes in the weights of the reward functions and the predefined threshold of easy and hard samples.
    - AdaCtrl is better at length control than prompt-based controllability.

**Strengths**

- The approach is simple and clearly explained.
- I appreciate the extent of the ablation studies on different components of the pipeline including the training data, the RL step, the reward functions and their hyperparameters.

**Weaknesses**

- I find that the result section is focusing more on the efficiency of AdaCtrl and less on its effectiveness. For example:
    - explicitly setting the difficulty tags in the controllability results in Table 2 has more effect on the length than on the accuracy. The easy mode results in much worse performance than the adaptive mode, while the hard mode does not improve much over the adaptive mode. This raises the question of the benefit of controllability to the accuracy.
    - In the controllability analysis in Figure 4(b), the paper focuses on response length comparison only. I would like to see the effect of prompt-based controllability on answer accuracy as well to better compare the trade-off.
- the study is missing quantitative comparisons to other SFT approaches (distilling concise paths from best-of-N sampling, omitting steps, compression, pruning steps) and RL approaches (length optimization through prompting, utility maximisation, heuristics) described in the related work section. It makes it hard to place this work within the literature.

**Audience:**

Yes

**Audience Explanation:**

It is a relevant paper to anyone working on dynamic reasoning with LLMs, controllability, and efficient SFT and RL algorithms for LLMs. However, as mentioned before, it is difficult to quantitatively compare it to other approaches that try to solve the same problem.

**Claims And Evidence:**

Yes

**Claims Explanation:**

Other that the balance between efficiency and effectiveness, the main claims in the paper are supported by exhaustive experiments.

**Requested Changes:**

###

- **Critical**
    - See weaknesses above.

- **recommended to strengthen the work**
    - besides Table1, it would be nice to present the accuracy-length trade-off in a Pareto frontier, as the numbers in the table are hard to read.

---

> ### Author Response · Authors · 2026-02-01
> **Response to Reviewer nv4F (1/2)**
>
> We sincerely thank you for your comprehensive and detailed evaluation, and we appreciate your acknowledgment of our extensive analysis and clear writing. We address your specific concerns below.
>
> Q1: Performance Trade-offs and Practical Value in Controllable Modes
>
> > I find that the result section is focusing more on the efficiency of AdaCtrl and less on its effectiveness. For example: explicitly setting the difficulty tags in the controllability results in Table 2 has more effect on the length than on the accuracy. The easy mode results in much worse performance than the adaptive mode, while the hard mode does not improve much over the adaptive mode. This raises the question of the benefit of controllability to the accuracy
>
> Regarding the performance trade-offs in controllable modes, we emphasize that the Easy mode is uniquely positioned as a specialized solution for resource-constrained scenarios. Its true value is best appreciated when compared directly to the Vanilla Qwen-2.5-Instruct baseline rather than solely against the Adaptive mode. In this comparison, the Easy mode maintains a comparable token efficiency (keeping responses extremely concise) while delivering superior robustness. For instance, on the challenging AIME 2024 benchmark, our Easy mode actually outperforms the Vanilla Qwen baseline (14.58% vs 11.25%) while maintaining a similarly compact length. Crucially, this represents a **10x reduction** in computational cost compared to standard reasoning models (which typically require thousands of tokens), yet it still yields higher accuracy than the original vanilla foundation, successfully validating its utility for latency-sensitive applications.
>
> Similarly, the Hard mode demonstrates superior effectiveness when compared to standard reasoning baselines. It consistently outperforms the heavy-RL baseline (R1-SFT-RL) on complex tasks (e.g., 22.08% vs 17.50% on AIME 2025) while often remaining more concise. The observation that the Adaptive mode performs nearly on par with this forced Hard mode serves as a strong validation of our gating mechanism, proving that the model autonomously identifies when to deploy maximum compute to match the Hard mode's ceiling. Ultimately, this controllability empowers users with a superior Pareto frontier: they can select Easy to beat Vanilla models at 1/10th the cost of reasoning models, or Hard to beat RL baselines at peak performance, with Adaptive bridging the best of both worlds.
>
> Q2: Accuracy Comparison for Prompt-based Controllability
>
> > In the controllability analysis in Figure 4(b), the paper focuses on response length comparison only. I would like to see the effect of prompt-based controllability on answer accuracy as well to better compare the trade-off.
>
> To address your concern, we have supplemented the accuracy data for the prompt-controlled baseline (S1.1-7B-prompt) and compared it directly with AdaCtrl's Easy and Adaptive modes. The results indicate that relying solely on prompts (e.g., "Please limit the length") not only fails in effective length control but also significantly underperforms AdaCtrl in terms of the efficiency-accuracy ratio.
>
> |                    | AIME24-Acc | AIME24-Len | AIME25-Acc | AIME25-Len | MATH500-Acc | MATH500-Len | GSM8K-Acc | GSM8K-Len |
> | ------------------ | ---------- | ---------- | ---------- | ---------- | ----------- | ----------- | --------- | --------- |
> | s1.1-7B-prompt     | 17.50      | 18026.17   | 16.67      | 16662.39   | 70.8        | 5360.42     | 90.83     | 1608.80   |
> | Adactrl (Easy)     | 14.58      | 1652.42    | 10.00      | 896.14     | 70.80       | 652.76      | 90.75     | 314.49    |
> | Adactrl (Adaptive) | 21.25      | 16889.50   | 19.17      | 15749.08   | 74.00       | 3195.69     | 90.98     | 349.34    |
>
> Specifically, on the simple GSM8K dataset, S1.1-7B-prompt achieved 90.83% accuracy but consumed an average of 1,608 tokens. In contrast, AdaCtrl (Easy) achieved comparable accuracy (90.75%) while compressing the length to 314 tokens, realizing a **5x efficiency improvement**. On the more challenging AIME2024, although S1.1-7B-prompt achieved slightly higher accuracy (17.50%) compared to AdaCtrl (Easy), its length skyrocketed to over 18,000 tokens, meaning the prompt completely failed to achieve the goal of "budget control." Conversely, AdaCtrl (Easy) successfully restricted the length to ~1,650 tokens, achieving over **10x compression**. Furthermore, when high performance is required, AdaCtrl (Adaptive) significantly outperforms the prompt baseline (21.25% vs 17.50% on AIME24) with similar token usage. This demonstrates that the prompt baseline essentially operates in an "uncontrolled" state, whereas AdaCtrl provides genuine budget control and a superior Pareto frontier. For this obervation, we have included the detailed accuracy comparison table in Appendix (*Efficiency-Accuracy Trade-off in Prompt-based Controllability*).

---

> ### Author Response · Authors · 2026-02-01
> **Response to Reviewer nv4F (2/2)**
>
> Q3: Quantitative Comparisons with Other SFT/RL Approaches
>
> > The study is missing quantitative comparisons to other SFT approaches... and RL approaches... It makes it hard to place this work within the literature.
>
> We greatly appreciate this point, to address this, we introduced a highly representative RL baseline: ARM [1]. This method represents the mainstream direction in related works that balances accuracy and length through "Utility Maximization."
>
> | Model       | AIME24         | AIME25         | MATH500       | GSM8K        |
> | ----------- | -------------- | -------------- | ------------- | ------------ |
> | AdaCtrl-7B  | 21.25/16889.50 | 19.17/15749.08 | 74.00/3195.69 | 90.98/349.34 |
> | ARM-7B      | –              | 16.70/3253     | 73.90/889     | 89.20/305    |
> | AdaCtrl-14B | 34.58/15173.21 | 25.83/14476.93 | 79.20/3209.84 | 94.09/316.12 |
> | ARM-14B     | –              | 20.00/3871     | 79.10/903     | 92.50/294    |
>
> We conducted comparative experiments at both the 7B and 14B scales. The results demonstrate that AdaCtrl exhibits a comprehensive advantage across the spectrum of difficulty. Notably, on simple tasks like GSM8K, AdaCtrl achieves **higher accuracy** than ARM (e.g., 90.98% vs 89.20% for 7B) while **maintaining a similarly concise efficiency**. However, the decisive advantage emerges on complex benchmarks. For instance, with the 7B model on the challenging AIME2025, AdaCtrl achieved an accuracy of **19.17%**, significantly outperforming ARM's 16.70%. Similarly, on the 14B model, AdaCtrl reached **25.83%**, surpassing ARM's 20.00%. This is primarily attributed to AdaCtrl's mechanism, which allows the model to fully expand long chains of thought on difficult problems (allocating ~15k tokens), whereas utility-based methods like ARM often suffer from "capability degradation" on hard problems due to excessive length penalization (stagnating around 3k tokens). This confirms that AdaCtrl effectively balances efficiency on simple tasks while possessing the unique ability to scale up reasoning compute to tackle tough problems, which is the core requirement for Large Reasoning Models. We have incorporated the performance of ARM into Table 1 and integrated the comparative analysis into the Main Results section of the revised manuscript.
>
> [1] ARM: Adaptive Reasoning Model
>
> Q4: Request for Pareto Frontier Plot
>
> > recommended to strengthen the work: besides Table1, it would be nice to present the accuracy-length trade-off in a Pareto frontier, as the numbers in the table are hard to read.
>
> Thank you for this valuable suggestion. We have visualized the Accuracy-Length Pareto Frontier on the AIME 2025 benchmark and included it as Figure 1 of the revised manuscript. This figure intuitively demonstrates that AdaCtrl constructs a dominant frontier situated to the upper-left of all baselines, highlighting its superior efficiency-effectiveness trade-off.

---

> ### Author Response · Authors · 2026-02-21
> **Follow up on the reviews**
>
> Dear Reviewer nv4F,
>
> Thank you again for your time and valuable feedback. We just wanted to gently follow up on our rebuttal and the revised PDF we submitted.
>
> We hope our responses, along with the updates incorporated into the manuscript, have effectively addressed your concerns. We remain fully available if any further clarification is needed before the discussion concludes.
>
> Best regards,
>
> The Authors

---

> > ### Comment · Reviewer_nv4F · 2026-03-02
> > **response**
> >
> > Thank you for the responses and the extra results. My concerns have been addressed.

---

> > > ### Author Response · Authors · 2026-03-02
> > >
> > > Thank you for the encouraging feedback. We’re pleased that our rebuttal and revisions have resolved your concerns. We also sincerely appreciate your thoughtful comments, which have helped strengthen our paper!

---

### Review · Reviewer_BBA3 · 2026-01-06

**Summary Of Contributions:**

These paper proposes a novel two-stage training paradigm designed to achieve both adaptive (model-driven) and controllable (user-driven) reasoning budgeting. Cold-start SFT aligns the model to generate concise or extensive responses based on prepended difficulty tags ([Easy] or [Hard]) using a curated mix of short and long CoT data. Difficulty-Aware RL utilizes GRPO with a specialized reward system, including a Difficulty Estimation Calibration Reward and a Difficulty-aware Length Reward. Experiments demonstrate that AdaCtrl can reduce response lengths while maintaining or even improving accuracy on math benchmarks like AIME2024 and GSM8K.

**strengths**
1. Cleverly uses the relative pass rates of multiple rollouts in GRPO as a "golden difficulty" signal to train the model's self-awareness.
2. Offers flexibility by allowing the model to self-assess difficulty while also providing a "human-in-the-loop" mechanism to explicitly steer the reasoning budget via tags.
3. Shows significant improvements over baselines.

**weaknesses**
1. The model must decide the difficulty at the very first token. This "judgment-before-thought" might fail on "trap" questions that appear easy but require deep reasoning to solve.

**Audience:**

Yes

**Audience Explanation:**

With the rise of reasoning LLMs, the researchers in LLMs and RL is highly focused on the efficiency of test-time scaling. AdaCtrl addresses a bottleneck of these models: the "overthinking" problem. The strategy of using RL to calibrate "model self-awareness" (difficulty estimation) is also a topic of significant interest.

**Claims And Evidence:**

Yes

**Claims Explanation:**

The paper provides strong empirical data across four major math benchmarks. The results clearly show that AdaCtrl achieves significant token reduction while maintaining or improving accuracy compared to SFT and RL baselines. Figure 4 and Figure 6 effectively visualize how the model learns to differentiate difficulty levels and adjust its "thinking budget" accordingly.

**Requested Changes:**

1. **Analysis of Reward Competition (Critical)**: There is a potential logical conflict between the Difficulty-aware Length Reward and the Difficulty Estimation Calibration Reward. Since length penalties are only applied when the model generates the [Easy] tag, the model has a strong incentive to "cheat" the system by labeling easy problems as [Hard] to bypass the length constraint and maximize reward. Please discuss whether this "reward hacking" (avoiding penalties by mislabeling) was observed during training.
2. **Baseline Prompting Fairness and Ablation (Critical)**: The current evaluation compares AdaCtrl (using a "Difficulty-Aware Prompt" that instructs the model to judge difficulty) against baselines using a standard "Baseline Prompt." It is well-documented in LLM literature that simply adding meta-cognitive instructions (like "judge the difficulty") can alter a model's activation paths and performance, regardless of the underlying RL training. Please conduct an additional experiment where the baseline is given a similar difficulty-aware prompt.

---

> ### Author Response · Authors · 2026-02-01
> **Response to Reviewer BBA3 (1/3)**
>
> We sincerely thank you for the highly constructive review. We appreciate your acknowledgment of our novel reward design using "golden difficulty" signals, the flexibility of our human-in-the-loop controllability, and the significant improvements demonstrated over baselines. We address your specific concerns below.
>
> Q1: Potential Failure of "Judgment-Before-Thought" on Trap Questions
>
> > The model must decide the difficulty at the very first token. This "judgment-before-thought" might fail on "trap" questions that appear easy but require deep reasoning to solve.
>
> Thank you for raising this insightful point regarding questions with deceptive surface simplicity. Regarding the hypothesis that "judgment-before-thought" might fail on such "trap questions," we emphasize that AdaCtrl's decision-making process is not driven by superficial textual features but is instead grounded in the **policy model's own verified capabilities**. The core design of our framework specifically handles this challenge through the **Difficulty Estimation Calibration Reward ($r_f$)**, which transforms the initial difficulty tag from a potential guess into a **calibrated capability forecast** derived from extensive training experience. In our design, if the model encounters a "trap question" that appears simple and incorrectly tags it as Easy, the resulting budget constraints will likely lead to failure on a task requiring deep reasoning. Crucially, this failure triggers a penalty in both the outcome reward ($r_o$) and the calibration reward ($r_f$). Throughout the Reinforcement Learning phase, these penalties compel the model to look beyond surface appearance and identify the latent structural features associated with failure, effectively learning that specific problem patterns require a Hard budget to maximize expected returns regardless of their apparent simplicity.
>
> To empirically validate that the model successfully identifies these complex scenarios, we evaluated its tagging accuracy by comparing its predicted tags against the "golden difficulty" determined by the actual pass rate of 16 parallel rollouts. As shown in the table below, AdaCtrl achieves remarkable precision across all benchmarks:
>
> | Tag Acc.    | AIME24 | AIME25 | MATH500 | GSM8K |
> | ----------- | ------ | ------ | ------- | ----- |
> | Adactrl-7B  | 93.33  | 94.58  | 87.82   | 96.58 |
> | Adactrl-14B | 92.08  | 95.42  | 89.47   | 96.40 |
>
> These high accuracy rates indicate that misclassifications are statistically rare, proving that the model successfully learns to anticipate the computational budget required for its own success even when facing deceptive questions. Furthermore, the training dynamics in **Figure 3** confirm this alignment, as the model autonomously converges to assigning Hard tags to the vast majority of challenging AIME problems while categorizing over 99% of simpler GSM8K problems as Easy. This demonstrates that after calibration, the "first-token commitment" functions as a robust and self-aware decision mechanism rather than a brittle heuristic. We have added a quantitative evaluation of Tagging Accuracy in the Appendix section of the revised manuscript.

---

> ### Author Response · Authors · 2026-02-01
> **Response to Reviewer BBA3 (2/3)**
>
> Q2: Analysis of Reward Competition and Potential "Reward Hacking"
>
> > Analysis of Reward Competition (Critical): There is a potential logical conflict between the Difficulty-aware Length Reward and the Difficulty Estimation Calibration Reward. Since length penalties are only applied when the model generates the Easy tag, the model has a strong incentive to "cheat" the system by labeling easy problems as Hard to bypass the length constraint and maximize reward. Please discuss whether this "reward hacking" (avoiding penalties by mislabeling) was observed during training
>
> Thanks for your constructive comment, we demonstrate that our multi-objective reward formulation effectively precludes this behavior through the counter-balancing force of the **Difficulty Estimation Calibration Reward ($r_f$)**. While the Difficulty-aware Length Reward ($r_l$) indeed applies constraints to the Easy mode, the $r_f$ component acts as a dominant corrective signal that aligns the model's tagging strategy with its actual success rate rather than speculative evasion. In our design, any attempt to "cheat" by labeling a clearly easy problem as Hard results in an immediate forfeiture of the $r_f$ reward, as the generated tag would contradict the "golden difficulty" derived from the GRPO rollout statistics. Our hyperparameter configuration ensures that the penalty for such misclassification consistently outweighs the marginal utility of escaping the length constraint, thereby making "honesty" the mathematically optimal strategy for maximizing expected returns.
>
> The training dynamics presented in **Figure 3** provide compelling empirical evidence that this theoretical hacking loop does not materialize in practice. On the simplest GSM8K dataset, we observed that the proportion of Easy tags rapidly increased and stabilized at over **99.85%**, rather than decreasing as would be expected if the model were attempting to game the system. Furthermore, this behavior is corroborated by the tagging accuracy results presented in **[A1]** (e.g., **96.58% accuracy** on GSM8K for AdaCtrl-7B). This exceptional precision confirms that the model is not arbitrarily converging to avoid penalties, but is accurately aligning its predictions with ground truth. This dual confirmation of high convergence to Easy tags on simple tasks combined with high prediction accuracy conclusively proves that the calibration reward successfully dominates the decision process, guiding the model toward an equilibrium of honest difficulty estimation and efficient budgeting rather than reward hacking.

---

> ### Author Response · Authors · 2026-02-01
> **Response to Reviewer BBA3 (3/3)**
>
> Q3: Baseline Prompting Fairness and Additional Experiment
>
> > Baseline Prompting Fairness and Ablation (Critical): The current evaluation compares AdaCtrl (using a "Difficulty-Aware Prompt" that instructs the model to judge difficulty) against baselines using a standard "Baseline Prompt." It is well-documented in LLM literature that simply adding meta-cognitive instructions (like "judge the difficulty") can alter a model's activation paths and performance, regardless of the underlying RL training. Please conduct an additional experiment where the baseline is given a similar difficulty-aware prompt.
>
> Thank you for this crucial suggestion to ensure a rigorous comparison. To decouple the effects of prompt engineering from our training methodology, we conducted the requested additional experiment. First, we would like to clarify that the "Baseline Prompt" used in our main evaluation was modeled after the official system prompt recommended for the Qwen-Math series. This ensures that the baseline model was evaluated under its optimal, intended setting to maximize its native performance. Following your suggestion, we applied the "Difficulty-Aware Prompt" (used in AdaCtrl) to this vanilla baseline to see if the performance gains were merely due to the prompt structure.
>
> |                                              | AIME24-Acc | AIME24-Len | AIME25-Acc | AIME25-Len | MATH500-Acc | MATH500-Len | GSM8K-Acc | GSM8K-Len |
> | -------------------------------------------- | ---------- | ---------- | ---------- | ---------- | ----------- | ----------- | --------- | --------- |
> | Qwen2.5-7B-Instruct-Baseline Prompt          | 11.25      | 1805.60    | 7.08       | 1174.06    | 73.00       | 628.91      | 91.58     | 272.93    |
> | Qwen2.5-7B-Instruct-Difficulty-Aware Prompt  | 10.42      | 1715.62    | 5.83       | 1402.52    | 67.6        | 538.29      | 87.34     | 237.83    |
> | Qwen2.5-14B-Instruct-Baseline Prompt         | 11.67      | 1043.20    | 10.42      | 1136.26    | 73.60       | 568.28      | 93.86     | 215.42    |
> | Qwen2.5-14B-Instruct-Difficulty-Aware Prompt | 4.58       | 830.68     | 5.0        | 686.01     | 73.0        | 409.86      | 85.75     | 188.47    |
>
> As observed, replacing the official baseline prompt with the difficulty-aware prompt leads to a noticeable drop in accuracy, exemplified by the 14B accuracy on AIME24 falling from 11.67% to 4.58%. We attribute this degradation to semantic interference, as the vanilla model lacks the internalized alignment to map these difficulty tags to appropriate reasoning budgets. Consequently, the extra instructions function as out-of-distribution noise that disrupts the model's standard reasoning path rather than serving as a valid control signal. This outcome explicitly validates the fairness of our original experimental setup, as it confirms that the standard "Baseline Prompt" represents the most robust and favorable configuration for the vanilla model. Therefore, the performance gap demonstrated in our paper is genuinely derived from our training methodology rather than an unfair prompting advantage. For this observation, we have added a dedicated section in Appendix (*Analysis of Baseline Prompting Fairness*) of the revised manuscript.

---

> ### Author Response · Authors · 2026-02-21
> **Follow up on the reviews**
>
> Dear Reviewer BBA3,
>
> Thank you again for your time and valuable feedback. We just wanted to gently follow up on our rebuttal and the revised PDF we submitted.
>
> We hope our responses, along with the updates incorporated into the manuscript, have effectively addressed your concerns. We remain fully available if any further clarification is needed before the discussion concludes.
>
> Best regards,
>
> The Authors

---

### Review · Reviewer_8nVw · 2026-01-20

**Summary Of Contributions:**

**Summary**
The paper introduces AdaCtrl, a framework designed to address the computational inefficiency and "overthinking" issues prevalent in Large Reasoning Models. The authors argue that while deep reasoning enhances performance on complex tasks, it incurs unnecessary overhead on simpler problems.

**Pros**
1. **Novel Reward Design:** Introduces a "Difficulty-Aware Length Reward" ($r_l$) that only penalizes verbosity when the problem is tagged as [Easy], preserving reasoning depth for hard tasks.
2. **Effective Pipeline:** Demonstrates that the "Cold-Start Fine-Tuning" stage is critical; without it, RL alone fails to learn efficient budgeting strategies.
3. **User Controllability:** Provides option for human control via explicit [Easy] and [Hard] tags, allowing users to manually prioritize efficiency or performance.

**Cons**
1. **Questionable Premise:** The motivation relies on "overthinking" phenomenon, which data suggests may be an artifact of distillation rather than an inherent flaw.
2. **Limited Evaluation:** Experiments are restricted to a single model family (Qwen), limiting generalizability.
3. **Performance Anomalies:** Unexplained degradation in forced "Hard" modes and inefficiencies compared to vanilla baselines on easy tasks.

**Audience:**

Yes

**Audience Explanation:**

The paper addresses the challenge of computational inefficiency in Large Reasoning Models, making it relevant to researchers focused on efficient inference and reinforcement learning.
Furthermore, the proposed framework for adaptive and controllable budgeting offers practical insights that would be valuable to people seeking to deploy reasoning models in resource-constrained environments.

**Claims And Evidence:**

Yes

**Claims Explanation:**

The paper presents a technically interesting solution to the problem of adaptive compute; however, there are several concerns regarding the framing of the problem and the robustness of the experimental analysis that need addressing.
1. **Misleading Premise Regarding "Overthinking" in Base Models:** The paper motivates the method by claiming LLMs suffer from substantial computational overhead due to overthinking. However, for example, Table 1 shows that the both vanilla base model is actually concise and highly-accuracte for some easier dataset already, including GSM8k and MATH. The "overthinking" issue (generating ~20k tokens) only appears in the distilled baselines (R1-SFT). Therefore, the paper is essentially targeting a problem introduced by the distillation process itself, rather than an inherent flaw in the base LLM.
2. **Lack of experiment from other model family:** The main result comes from Qwen models only, and it should contain other family of models for stating to solve a broadly exist problem
3. **Potential "Mode Collapse" at Inference Time:** The framework requires the model to commit to a difficulty tag ([Easy] or [Hard]) at the very first token of generation. The paper does not analyze the recoverability of this decision. If the model incorrectly tags a hard problem as [Easy], it is effectively locked into a concise budget and likely fails. An analysis of "False Negatives" (Hard problems tagged Easy) is missing to understand the failure modes of this rigid tagging system.
4. **Performance Degradation in Forced "Hard" Mode:** Table 2 reveals that on the MATH500 dataset, the forced [Hard] mode achieves lower accuracy (71.20%) than the Adaptive mode (74.00%). Intuitively, allocating more budget should maintain or improve performance unless the reasoning quality degrades. The author does not provide a clear explanation on this. Furthermore, the [Easy] mode has 1: lower accuracy, 2: higher average token usage than the vanilla mode. I think an explanation is also necessary in this part.

**Requested Changes:**

The author should address concerns mentioned in "Cons" section above, and answer the following questions, for strengthen the work.
**Questions**
1. Base on my use experience, the vanilla qwen model is actually not affected by overthinking, and this is also reflected in Table 1. I think the author need to compare the response length of the common correct subset between the vanilla Qwen model and other method, including AdaCtrl.
2. Why is the average response length of AdaCtrl in forced *Hard* mode noticeably shorter (~20%) than the R1-SFT and R1-SFT-RL baselines, despite both being trained on similar 'hard' reasoning trajectories? Does the inclusion of *Easy* samples in the training data cause a 'conciseness transfer' that regularizes the lengthy reasoning patterns even when the *Hard* tag is active? If so, does this reduction in length come at the cost of missing specific reasoning steps present in the pure R1 baselines?
3. Models like deepseek-distilled-qwen-7b is actually suffered from overthinking and backtracking a lot. Can this method use to mitigate this phenomenon for a heavy RLed model?

---

> ### Author Response · Authors · 2026-02-01
> **Response to Reviewer 8nVw (1/5)**
>
> We sincerely thank you for the in-depth review, and appreciate your acknowledgment of our novel reward design, effective pipeline, and the practical value of user controllability. We will address each of your concerns below.
>
> Q1: Misleading Premise Regarding "Overthinking" in Base Models
>
> > The paper motivates the method by claiming LLMs suffer from substantial computational overhead due to overthinking. However, for example, Table 1 shows that the both vanilla base model is actually concise and highly-accuracte for some easier dataset already, including GSM8k and MATH. The "overthinking" issue (generating ~20k tokens) only appears in the distilled baselines (R1-SFT). Therefore, the paper is essentially targeting a problem introduced by the distillation process itself, rather than an inherent flaw in the base LLM.
>
> A1: Thank you for the opportunity to clarify the scope of our motivation. We fully acknowledge that vanilla instruct models are naturally concise on simple tasks. However, the primary objective of our work is not to critique base models, but to address the **uncontrolled growth of generation length** that inevitably emerges when models undergo Reinforcement Learning (RL) to push the boundaries of reasoning performance. Even setting aside the specific effects of distillation, when models optimize for complex reasoning tasks, they often discover that generating longer and more redundant reasoning paths can speculatively maximize accuracy rewards [1]. This phenomenon leads to significant computational overhead even on straightforward queries where such depth is unnecessary. As evidenced by Figure 6 in our paper, we observed a rapid and uncontrolled increase in average response length across *all* datasets during the early stages of RL training. AdaCtrl’s unique contribution lies in its ability to actively calibrate and suppress this unnecessary expansion during RL iterations via the difficulty-aware length reward ($r_l$). This ensures that the model attains deep reasoning capabilities while preserving the computational efficiency typical of vanilla models for simpler problems—a balance that standard RL pipelines fail to maintain. We have updated the Abstract and Introduction (Paragraph 1) in the revised manuscript to explicitly clarify this scope.
>
> [1] DeepSeek-R1: Incentivizing Reasoning Capability in LLMs via Reinforcement Learning
>
> Q2: Lack of experiment from other model family
>
> > The main result comes from Qwen models only, and it should contain other family of models for stating to solve a broadly exist problem
>
> Thank you for the constructive suggestion. We fully agree on the importance of demonstrating generalizability across model families. To address this, we have extended our evaluation to include the **Llama-3.1-8B-Instruct** model. The preliminary results are presented below:
>
> |                                     | AIME24-Acc | AIME24-Len | AIME25-Acc | AIME25-Len | MATH500-Acc | MATH500-Len | GSM8K-Acc | GSM8K-Len |
> | ----------------------------------- | ---------- | ---------- | ---------- | ---------- | ----------- | ----------- | --------- | --------- |
> | LLama3.1-8B-Instruct                | 3.58       | 10901.07   | 2.08       | 10383.25   | 46.8        | 4203.35     | 85.14     | 1319      |
> | LLama3.1-8B-Instruct-R1-SFT         | 3.33       | 19595.04   | 3.33       | 19040.31   | 48.4        | 8792.70     | 83.78     | 3387.50   |
> | LLama3.1-8B-Instruct-Cold-Start-SFT | 2.92       | 15811.69   | 2.08       | 17247.47   | 43.2        | 5327.22     | 83.02     | 641.50    |
> | LLama3.1-8B-Instruct-R1-SFT-RL      | 7.08       | 19377.87   | 5.83       | 18026.00   | 53.2        | 11140.40    | 83.09     | 4371.49   |
> | LLama3.1-8B-Instruct-Adactrl        | 9.17       | 16630.41   | 6.25       | 16364.67   | 55.6        | 6397.70     | 87.49     | 881.69    |
>
> As shown in the table, AdaCtrl consistently demonstrates its adaptive length control advantages on the Llama architecture. Compared to the `R1-SFT-RL` baseline, AdaCtrl achieves higher accuracy (e.g., 9.17% vs 7.08% on AIME24) while significantly reducing token consumption on simpler tasks (e.g., 881.69 vs 4371.49 on GSM8K). Although the absolute performance is lower than the Qwen-based models (likely due to the differing mathematical capabilities of the base models), these results confirm that our two-stage training pipeline effectively enables difficulty-awareness and budget control across different model architectures, exhibiting strong transferability.
>
> We have incorporated these experimental results and the corresponding analysis into the Analysis section (New Subsection: Generalizability across Model Families) of the revised manuscript.

---

> ### Author Response · Authors · 2026-02-01
> **Response to Reviewer 8nVw (2/5)**
>
> Q3: Potential "Mode Collapse" at Inference Time and False Negatives
>
> > The framework requires the model to commit to a difficulty tag (Easy or Hard) at the very first token of generation... If the model incorrectly tags a hard problem as Easy, it is effectively locked into a concise budget and likely fails. An analysis of "False Negatives" (Hard problems tagged Easy) is missing to understand the failure modes of this rigid tagging system.
>
> Thank you for the insightful question regarding the risks of first-token decision-making. We would like to clarify that our definitions of "Easy" and "Hard" are grounded in the **policy model's own capabilities**, rather than human perception. Our goal is for the model to predict the optimal budget based on its own proficiency. We acknowledge that for an uncalibrated model, committing to a budget upfront carries the risk of underestimation (False Negatives). However, the core design of AdaCtrl specifically mitigates this through the **Difficulty Estimation Calibration Reward ($r_f$)**, which transforms the difficulty tag from a "rigid guess" into a **calibrated capability forecast**. During the RL phase, the $r_f$ reward incentivizes the model to align its difficulty prediction with its **actual success rate** derived from multiple GRPO rollouts (the "post-hoc golden difficulty"). Essentially, if the model incorrectly tags a hard problem as Easy and fails due to an insufficient budget, the RL penalty forces it to correct this behavior and assign a Hard tag in subsequent iterations. To quantify the effectiveness of this calibration, we evaluated the model's tagging accuracy by comparing its predicted tags against the ground-truth difficulty (determined by the pass rate of 16 parallel rollouts),  Specifically, a problem was classified as 'Easy' if solved correctly more than 10 times (matching our training threshold), and 'Hard' otherwise. As shown in the table below, AdaCtrl achieves remarkable precision across all benchmarks:
>
> | Tag Acc.    | AIME24 | AIME25 | MATH500 | GSM8K |
> | ----------- | ------ | ------ | ------- | ----- |
> | Adactrl-7B  | 93.33  | 94.58  | 87.82   | 96.58 |
> | Adactrl-14B | 92.08  | 95.42  | 89.47   | 96.40 |
>
> These high accuracy rates indicate that "False Negatives" are statistically rare, proving that the model successfully learns to anticipate the computational budget required for its own success. Furthermore, the training dynamics in **Figure 3** provides visual confirmation of this alignment: the model autonomously converges to assigning Hard tags to the vast majority of challenging AIME problems while categorizing over 99% of simpler GSM8K problems as Easy. This demonstrates that after $r_f$ calibration, the model develops the necessary self-awareness to mitigate mislabeling risks, and the "first-token commitment" effectively functions as a robust, self-aware decision mechanism rather than a brittle constraint.  We have added a quantitative evaluation of Tagging Accuracy in the Appendix section of the revised manuscript.

---

> ### Author Response · Authors · 2026-02-01
> **Response to Reviewer 8nVw (3/5)**
>
> Q4: Performance Degradation in Forced "Hard" Mode
>
> >  Table 2 reveals that on the MATH500 dataset, the forced Hard mode achieves lower accuracy (71.20%) than the Adaptive mode (74.00%) ... Furthermore, the Easy mode has 1: lower accuracy, 2: higher average token usage than the vanilla mode. I think an explanation is also necessary in this part.
>
> Thank you for highlighting these nuances. Regarding the performance on MATH500, the lower accuracy of the forced Hard mode compared to the Adaptive mode is indeed an important observation. Our analysis reveals that this performance drop stems from the subset of problems where the Adaptive mode correctly identified that a concise solution (Easy tag) was sufficient, but the forced Hard mode introduced unnecessary complexity.*
>
> To quantify this, we isolated the samples where the **Adaptive mode autonomously selected the `[Easy]` tag** and compared their outcomes against when the same samples were **forced into `[Hard]` mode**. As summarized in the table below, forcing a long-context budget on these problems results in a net accuracy loss:
>
> | Adaptove Mode $\to$ Hard Mode | **Gain** (Easy $\times$ $\to$ Hard $\checkmark$) | **Loss** (Easy $\checkmark$ $\to$ Hard $\times$) | Overthinking Analysis within "Loss"                          |
> | ----------------------------- | ------------------------------------------------ | ------------------------------------------------ | ------------------------------------------------------------ |
> | Adactrl-7B                    | 27 samples                                       | 47 samples                                       | **32 / 47** cases initially derived the correct answer but drifted into error due to forced verbosity. |
> | Adactrl-14B                   | 18 samples                                       | 38 samples                                       | **27 / 38** cases failed due to similar redundancy-induced errors. |
>
> As illustrated in the table, the forced Hard mode indeed demonstrates the potential to raise the performance ceiling, as evidenced by the 27 'Gain' samples where extended reasoning successfully solved complex problems that failed in the concise mode. However, this benefit is currently outweighed by the 47 'Loss' cases. Crucially, our error analysis reveals that these failures do not stem from a lack of capability. Taking AdaCtrl-7B as an example, through human validation, in 32 out of the 47 'Loss' cases (about **68%**), the model had initially derived the correct answer but was compelled to drift into incorrect revisions due to the forced requirement for extended reasoning (i.e., "forced verbosity"). This indicates that the "intrinsic capability" of the Hard mode is actually higher than its final accuracy suggests, but it is compromised by "thinking noise" such as unnecessary self-doubt or hallucinatory verification, on simpler queries. This finding strongly validates the design of AdaCtrl’s Adaptive mode: it successfully captures the high-ceiling benefits (the 'Gain' cases) while intelligently terminating reasoning early to prevent the 'Loss' cases where overthinking creates errors. Crucially, however, this dynamic shifts entirely on genuinely challenging benchmarks like AIME, where the deep reasoning capability of the Hard mode is indispensable and leads to significant performance gains (e.g., boosting 7B accuracy to **22.08%**). This stark contrast underscores the critical value of AdaCtrl’s Adaptive mode: it intelligently navigates this trade-off, successfully mitigating the overthinking noise while preserving the capacity to deploy extensive reasoning budgets when truly needed for complex problems.
>
> Regarding the comparison between the Easy mode and the vanilla model, we emphasize that our method maintains **comparable efficiency and performance** on simple tasks while delivering **superior robustness** on challenging ones. On GSM8K, the response length is remarkably similar (314 vs 273 tokens) with competitive accuracy (90.75% vs 91.58%). However, on complex benchmarks where vanilla models typically struggle, our Easy mode actually **outperforms the vanilla baseline**: for instance, on AIME 2024, it achieves **14.58% accuracy compared to Vanilla's 11.25%**, and on AIME 2025, it reaches **10.00% vs 7.08%**.  Crucially, this highlights the core value of our framework: it offers the flexibility to secure competitive results with merely **1/10th of the standard reasoning budget** (via Easy mode), while retaining the capacity to dynamically expand reasoning length by **10x** to unlock maximum performance. This wide dynamic range allows AdaCtrl to navigate the performance-cost trade-off more effectively than any static baseline. We have added the detailed breakdown of the "Gain/Loss" analysis on MATH500 and the specific comparison between Easy Mode and Vanilla baselines in Appendix.
>
> [1] Between Underthinking and Overthinking: An Empirical Study of Reasoning Length and Correctness in LLMs

---

> ### Author Response · Authors · 2026-02-01
> **Response to Reviewer 8nVw (4/5)**
>
> Q5: Common Correct Subset Length and "Conciseness Transfer"
>
> > Base on my use experience, the vanilla qwen model is actually not affected by overthinking, and this is also reflected in Table 1. I think the author need to compare the response length of the common correct subset between the vanilla Qwen model and other method, including AdaCtrl.
>
> Thank you for this suggestion. We fully agree that Vanilla Qwen functions efficiently as a System 1 model on simple tasks.However, as clarified in **[A1]**, our primary focus is not on addressing inherent flaws in vanilla models, but specifically on curbing the **uncontrolled and redundant length growth** that inevitably emerges during the **Reinforcement Learning (RL)** process for reasoning-enhanced models. To directly address your request, we compared the average response lengths on the **GSM8K Common Correct Subset** for Vanilla Qwen, `R1-SFT-RL`, and AdaCtrl. The results are compelling:
>
> |      | Common Correct Subset Size | Vanilla Qwen | AdaCtrl | R1-SFT-RL |
> | ---- | -------------------------- | ------------ | ------- | --------- |
> | 7B   | 1084 samples               | 295.0        | 307.0   | 2043.0    |
> | 14B  | 1154 samples               | 283.0        | 285.0   | 2106.0    |
>
> These results show that AdaCtrl achieves lengths **remarkably close** to Vanilla Qwen (only ~12 tokens difference for 7B, and nearly identical for 14B), effectively taming the exponential length growth seen in the `R1-SFT-RL` baseline (which is ~7x longer). This demonstrates that AdaCtrl successfully bridges the gap: it replicates the efficiency of a base model on simple tasks while retaining the high-performance reasoning capabilities of RL-tuned models.
>
> Q6: Shorter length of AdaCtrl
>
> > Why is the average response length of AdaCtrl in forced Hard mode noticeably shorter (~20%) than the R1-SFT and R1-SFT-RL baselines, despite both being trained on similar 'hard' reasoning trajectories? Does the inclusion of Easy samples in the training data cause a 'conciseness transfer' that regularizes the lengthy reasoning patterns even when the Hard tag is active? If so, does this reduction in length come at the cost of missing specific reasoning steps present in the pure R1 baselines?
>
> Regarding the noticeable conciseness of AdaCtrl-Hard mode compared to R1-based baselines (especially the heavy-RL baseline `R1-SFT-RL`), we agree that this indeed stems from the "conciseness transfer" effect introduced by our mixed-data cold-start strategy. Crucially, this reduction does **not** come at the cost of missing reasoning steps; rather, it eliminates non-functional redundancies.
>
> Experimental evidence supports this: AdaCtrl-Hard consistently outperforms the longer `R1-SFT-RL` baseline. For example, on MATH500, AdaCtrl-Hard achieves **71.20% accuracy with ~5,960 tokens**, whereas `R1-SFT-RL` achieves only **66.80% with ~8,421 tokens**. Similarly, on AIME2025, AdaCtrl outperforms the baseline (**22.08% vs 17.50%**) while being more concise. To further verify reasoning quality, we analyzed 100 random reasoning traces from both `R1-SFT-RL` and AdaCtrl-Hard using `Gemini-3-Pro`. We found that while `R1-SFT-RL` exhibits 22 distinct reasoning patterns, AdaCtrl exhibits **28 patterns**, covering all of R1's patterns (e.g., "Decomposition", "Backtracking") and demonstrating **new emergent patterns** such as "Algorithmic Thinking", "Heuristic Reasoning", and "Case Analysis". This indicates that our method produces denser, higher-quality reasoning chains, effectively "purifying" the logic rather than truncating it. We have added a new subsection "Qualitative Analysis of Reasoning Patterns" and  Figure 6 (Word Cloud of Reasoning Patterns) in the Analysis section of the revised manuscript.

---

> ### Author Response · Authors · 2026-02-01
> **Response to Reviewer 8nVw (5/5)**
>
> Q7: Mitigating Overthinking in Heavy RL Models
>
> > Models like deepseek-distilled-qwen-7b is actually suffered from overthinking and backtracking a lot. Can this method use to mitigate this phenomenon for a heavy RLed model?
>
> Thank you for this practical question. We have conducted exploratory experiments applying our method to these heavy-RLed models during our preliminary exploratory phase. However, we found that during the initial Supervised Fine-Tuning (SFT) phase, which is necessary to introduce difficulty-aware tags, the imposition of length constraints led to severe **"Capability Collapse."** Our analysis reveals that models trained with heavy RL converge to a local optimum where they develop a strong path dependency on lengthy, redundant reasoning paths. Disrupting this structure via SFT breaks their internal logical coherence, resulting in a sharp and often irreversible degradation in performance. Crucially, we observed that even when we proceeded to the subsequent Reinforcement Learning (RL) phase, the model failed to recover its original reasoning capabilities.
>
> This finding underscores that "overthinking" is a structural pathology formed during the RL process, making it difficult to fix with a post-hoc application of our framework. Consequently, the core value of AdaCtrl lies in offering a **preventative training paradigm**: by integrating difficulty-aware tags and dynamic budget rewards ($r_l$) throughout the *entire* lifecycle (from Cold-Start to RL), AdaCtrl prevents the formation of invalid redundancy from the outset. As evidenced by our results, this approach successfully maintains high accuracy (e.g., 90.98% on GSM8K) while keeping the response length minimal (349 tokens), achieving a balance that post-hoc adaptation fails to reach.

---

> > ### Comment · Reviewer_8nVw · 2026-02-02
> >
> > Thanks for the modifcation and clarification for all of my questions and concerns. I think you have convincing and clear evidence for your claim now.

---

> > > ### Author Response · Authors · 2026-02-02
> > > **Appreciation for your constructive comments**
> > >
> > > Thank you for your positive feedback! We are glad that our rebuttal and revisions address your concerns. Thanks again for your insightful comments that help improve our work greatly.

---

### Decision · Action_Editor_oPG1 · 2026-03-13

**Recommendation:** Accept as is

**Additional Comments:**

This is an interesting paper on controlling reasoning in large language models. The basic idea is to adapt the quantity of reasoning based on criteria like difficulty and user-selected abilities. The concept is good and particularly timely given the importance of reasoning models (and the computational expense of heavy reasoning). The authors' results are strong.

The reviewers are in favor of the paper; the authors did a nice job answering their initial questions. Based on this I recommend acceptance.

**Audience:**

Yes

**Audience Explanation:**

This is in a very important area for language models today, so definitely fits the bill.

**Claims And Evidence:**

Yes

**Claims Explanation:**

The paper provides empirical evidence for all of its claims.